# Memory effect assisted imaging through multimode optical fibres

Shuhui Li [1,2✉], Simon A. R. Horsley [1], Tomáš Tyc[3,4], Tomáš Čižmár [4,5] & David B. Phillips [1✉]

When light propagates through opaque material, the spatial information it holds becomes scrambled, but not necessarily lost. Two classes of techniques have emerged to recover this information: methods relying on optical memory effects, and transmission matrix (TM) approaches. Here we develop a general framework describing the nature of memory effects in structures of arbitrary geometry. We show how this framework, when combined with wavefront shaping driven by feedback from a guide-star, enables estimation of the TM of any such system. This highlights that guide-star assisted imaging is possible regardless of the type of memory effect a scatterer exhibits. We apply this concept to multimode fibres (MMFs) and identify a 'quasi-radial' memory effect. This allows the TM of an MMF to be approximated from only one end - an important step for micro-endoscopy. Our work broadens the applications of memory effects to a range of novel imaging and optical communication scenarios.

[1] Physics and Astronomy, University of Exeter, Exeter, UK. [2] Wuhan National Laboratory for Optoelectronics, School of Optical and Electronic Information, Huazhong University of Science and Technology, Wuhan, China. [3] Department of Theoretical Physics and Astrophysics, Faculty of Science, Masaryk University, Brno, Czech Republic. [4] Institute of Scientific Instruments of CAS, Brno, Czech Republic. [5] Leibniz Institute of Photonic Technology, Jena, Germany. ✉email: shli@hust.edu.cn; d.phillips@exeter.ac.uk

 

Coherent images are encoded in spatial light modes: patterns in the intensity and phase of light. When these patterns propagate through an opaque scattering material, such as frosted glass, a multimode optical fibre (MMF), or biological tissue, they distort and fragment, and the spatial information they carry becomes scrambled. This corrupts the formation of images of objects hidden behind or inside turbid media. In the last decade or so, a series of pioneering studies have demonstrated how digital light shaping technology can be used to measure and reverse scattering effects - unscrambling the light back to the state it was in before it entered the medium[1–4]. These techniques take advantage of the linear (in electric field) and deterministic nature of the scattering, and have enabled focussing and imaging through scattering systems[5–9]. At the heart of this capability lies the transmission matrix (TM) concept, which describes the scattering as a linear operation relating a set of input spatial light modes incident on one side of the scatterer, to a new set of output modes leaving on the opposite side[3]. Once the TM of a scatterer has been measured, it tells us what input field is required to create an arbitrary output field, and thus permits the transmission of images through the scatterer. However, measurement of the TM typically requires full optical access to both sides of the scatterer - impossible when the image plane is, for example, embedded in living tissue.

To overcome this limitation and look inside scattering environments, a useful suite of tools has emerged in the form of optical memory effects: the presence of underlying correlations between the incident and transmitted fields[10–13]. These hidden correlations have been extensively studied in thin randomly scattering layers which have direct applications to imaging through biological tissue. In this case tilting[10], shifting[12], spectral[14] and temporal[15] correlations have been revealed. The tilt and shift memory effects are related to correlations in the Fourier-space or real-space TM of a scatterer - and have enabled extraction of subsets of the TM which can be used to image over small areas inside scattering systems.

In parallel with these advances, TM approaches have also spurred the development of alternative methods of seeing into tissue much more deeply, albeit invasively, by guiding light along narrow waveguides. These techniques use MMFs to achieve high-resolution imaging at the tip of a needle - acting as ultra-low footprint endoscopes. Modal dispersion scrambles coherent optical signals transmitted through MMFs, and so before they can be deployed in scanning imaging systems, their TM must also first be measured. MMF based micro-endoscopy has great potential for deep tissue imaging, as indicated by a swathe of recent successes[16–19], yet a major challenge holding back broader uptake of this technique is the fragility of the TM used to control the optical field at the distal (far) facet. After TM calibration, which as described above conventionally requires access to both ends of the fibre, the MMF must be held completely static, as even small perturbations in fibre configuration (e.g., bends or twists) or temperature de-phase the propagating fibre modes, severely reducing the contrast of focussed spots at the distal facet, and the fidelity of the reconstructed images. Flexible operation of micro-endoscopic imaging systems based on current optical fibre technology require a TM characterisation method that can be rapidly performed on the fibre in-situ, with access only at the proximal (near) end.

Here we develop a general framework describing how monochromatic memory effects arise in samples of arbitrary geometry, and apply it to the MMF case. We show that deployment of a guide-star located on the distal facet of an MMF (and capable of reporting its local field intensity to the proximal end), combined with an estimate of the basis in which the TM is close to diagonal, provides a way to approximate the TM of, and thus image

through, optical fibres. Crucially, this approach only requires access to the proximal end of the MMF: offering a route to in-situ TM calibration of flexible micro-endoscopic imaging systems. More generally, the concept we describe here enables guide-star based scanning imaging to be performed through scattering systems of any geometry, without invoking conventional shift or tilt memory effects, as long as we have an estimate of the basis in which the TM is quasi-diagonal. Our work broadens the applications of memory effects beyond thin randomly scattering layers to a range of novel imaging and optical communication scenarios.

## Results

**The memory effect in an arbitrary basis.** We start by describing the emergence of memory effects in optical systems of arbitrary geometry. These systems may encompass, for example, the well-understood case of thin scattering layers, but also networks of wave-guides, or any other linear optical system through which light is transmitted. We assume that we know the geometry of the object before us - for example a thin scattering layer, or an optical fibre - but we do not have detailed knowledge of its scattering properties - for example we do not know the spatial function of the refractive index throughout a layer, or the length or bend configuration of a fibre. Given this level of prior information, we describe when and how optical memory effects may be used to achieve imaging.

In the general case, our aim is to image through a scattering system that has an unknown monochromatic TM, $\mathbf{T}$. As a first step, let's assume that the geometry of the system reveals enough information for us to estimate a basis in which $\mathbf{T}$ is *approximately* diagonal, i.e., $\mathbf{T} = \mathbf{UDU}^{-1}$, where we know matrix $\mathbf{U}^{-1}$ which allows us to transform to the quasi-diagonal basis from knowledge of the field in real-space. We assume that $\mathbf{D}$ is quasi-diagonal (i.e., it has a significant proportion of its power concentrated on the diagonal), but we have no knowledge of the complex elements of $\mathbf{D}$. We consider the following questions: is advance knowledge of a basis in which the TM of a scatterer is quasi-diagonal enough information to (i) predict a memory effect in, and (ii) image through, the system?

The memory effect occurs when a given modulation (i.e., transformation) of the field incident onto a scatterer modifies the output field in a deterministic way - for example, creating a lateral displacement of the output field. To explore this idea in a general basis, we express the input field as a column vector $\mathbf{u}$, which holds the complex coefficients of the input field in real-space. Incident field $\mathbf{u}$ is transformed into an output field $\mathbf{v} = \mathbf{Tu}$ via propagation through a scatterer. We consider a modulation of the incident field in the form $\mathbf{u}' = \mathbf{Ou} = \mathbf{UMU}^{-1}\mathbf{u}$, where $\mathbf{M}$ is a diagonal matrix. After such a modulation, the output field $\mathbf{v}$ is changed to $\mathbf{v}'$, which can be written as

$$\mathbf{v}' = \mathbf{Tu}' = \mathbf{UDMU}^{-1}\mathbf{u}. \tag{1}$$

Given that $\mathbf{M}$ is exactly diagonal, and $\mathbf{D}$ is close to diagonal, the two matrices will commute with an error $[\mathbf{D}, \mathbf{M}]$ that is due to the off-diagonal elements of $\mathbf{D}$, i.e., $\mathbf{DM} = \mathbf{MD} + [\mathbf{D}, \mathbf{M}]$. As $\mathbf{D}$ becomes closer to diagonal this error will reduce, becoming zero in the limit of an exactly diagonal matrix. We therefore rewrite Eq. (1) as

$$\begin{aligned} \mathbf{v}' &= \mathbf{UMDU}^{-1}\mathbf{u} + \mathbf{U}(\mathbf{DM} - \mathbf{MD})\mathbf{U}^{-1}\mathbf{u} \\ &= \mathbf{OTu} + \mathbf{U}[\mathbf{D}, \mathbf{M}]\mathbf{U}^{-1}\mathbf{u}, \end{aligned} \tag{2}$$

which shows that for input transformations of the form $\mathbf{O} = \mathbf{UMU}^{-1}$, the output field undergoes the same transformation with an error $\boldsymbol{\eta} = \mathbf{U}[\mathbf{D}, \mathbf{M}]\mathbf{U}^{-1}\mathbf{u}$. In more condensed notation,

Eq. (2) is

$$\mathbf{v}' = \mathbf{Ov} + \boldsymbol{\eta} \sim \mathbf{Ov}. \qquad (3)$$

Equation (3) describes a general form of the memory effect. In answer to our first question (i), Eq. (3) tells us that without knowing anything about the elements of the quasi-diagonal matrix $\mathbf{D}$, we can deterministically modify the field transmitted through a scattering system. The basis in which the TM is diagonal is linked to the form of the field modification, i.e., memory effect, that is possible. This highlights the existence of an infinite family of memory effects operating in scattering systems of different geometries, each requiring its own unique but predictable set of transformations on input fields, with a deterministic transformation of the output that is not, in general, a translation of the field. We now focus on some special cases, and highlight when imaging is possible using these memory effects.

**Memory effect based imaging through scattering systems**. We first consider the well-known tilt–tilt memory effect[10,11]. In a thin randomly scattering layer, multiple scattering is governed at a statistical level by a diffusion process, meaning that light focussed to a given lateral position on the input will only diffuse locally to nearby lateral positions at the output. In this case our assumption is that $\mathbf{U} = \mathbf{I}$, the identity matrix, i.e., the monochromatic TM is quasi-diagonal in the position basis (and equivalently exhibits diagonal correlations in the Fourier basis[12]). Our transformation $\mathbf{O}$ is thus also diagonal in the position basis: $\mathbf{O} = \mathbf{IMI} = \mathbf{M}$. This means that any spatially varying phase change applied to the input field, is also applied to the output field. More specifically, a tilt of the input beam results in a tilt of the output beam, in which case the diagonal elements of $\mathbf{M}$ correspond to a phase ramp modulation given by $\mathbf{M} = \text{diag}[e^{i(\delta k_x \mathbf{x} + \delta k_y \mathbf{y})}]$, where $\text{diag}[\mathbf{d}]$ places vector $\mathbf{d}$ on the main diagonal of a square zero matrix, $\mathbf{x}$ and $\mathbf{y}$ are vectors specifying the $x$ and $y$ coordinates of each pixel on the input plane, and $\delta k_x$ and $\delta k_y$ specify the desired change in $x$- and $y$-components of the wave-vector normal to the tilted wavefront.

Tilting of the output beam is not in itself useful to image the output plane of the sample. However, in the far-field of the output, this tilt corresponds to a lateral shift, and so unknown speckle patterns can be controllably translated in the far-field. Bertolotti et al.[20] showed that imaging is then possible from the input side by measuring the fluorescent intensity excited by these unknown patterns as they are scanned in two dimensions. This procedure yields the amplitudes of the Fourier components of the image, but not the phases of these Fourier components. Despite this missing information, a diffraction limited image of an object in the far-field can be estimated using a phase retrieval algorithm with appropriate constraints[21].

Next we consider how the more recently discovered shift–shift memory effect appears in our general framework[12]. The shift–shift memory effect occurs for some anisotropic disordered materials, assuming that the scatterer has a thickness less than the transport mean-free-path (TMFP) of the system. The TMFP is the average length over which scattering randomises the photon direction (wave-vector). This means that we expect only small changes in the angular deviation of rays from input to output, and so assume the TM of the scatterer to be quasi-diagonal when represented in the two dimensional Fourier basis $\mathbf{U} = \mathbf{F}^{-1}$. In this case a transformation $\mathbf{O} = \mathbf{F}^{-1}\mathbf{MF}$ of the input beam leads to approximately the same transformation of the output beam. Again choosing a phase ramp along the diagonal of $\mathbf{M}$, $\mathbf{O}$ becomes a spatial shift operator, and so lateral displacement of the input beam results in a corresponding shift of the output beam, i.e., $\mathbf{M} = \text{diag}[e^{-i(\mathbf{k}_x \delta x + \mathbf{k}_y \delta y)}]$, where $\mathbf{k}_x$ and $\mathbf{k}_y$ are vectors specifying the $x$- and $y$-components of the wave-vectors arriving at the input plane, and $\delta x$ and $\delta y$ specify the desired lateral shift of the input . Equivalently the TM exhibits diagonal correlations in real-space[12]. In this case, imaging in the near-field of the output is theoretically possible by once again scanning unknown speckle patterns in 2D and performing phase retrieval[20]. We note that the shift–shift and tilt–tilt memory effects are Fourier duals of each other, one moving the field, the other re-directing it. As a Fourier transform and its inverse contain factors of $\exp(-ikx)$ and $\exp(+ikx)$ respectively, then the applied phases in the two memory effects must similarly differ by a minus sign.

The addition of a 'guide-star' embedded in the scatterer provides more information, and so improves the memory effect based imaging that is possible: yielding images with higher a signal-to-noise ratio (SNR), and without the need for phase-retrieval. First implemented in astronomy to correct for atmospheric turbulence[22], a guide-star is a point within the scatterer capable of signalling the intensity of its local field to the outside world. For use in microscopy, guide-stars have been created from highly reflective embedded particles[23], or via a range of alternative methods exploiting, for example, fluorescent[24], magnetic[25], movement-based[26], photo-acoustic[27], or acousto-optic[28] properties. The input field required to focus on a guide-star, $\mathbf{u}^{gs}$, can be calculated by phase conjugation of a field emanating from the guide-star[2], or by phase stepping holography[4]. $\mathbf{u}^{gs}$ constitutes a single column of the inverse of the TM of the scatterer, $\mathbf{T}^{-1}$ (with the output discretised in the real-space 'pixel' basis). In cases where the TM is unitary (or at least approximately so), $\mathbf{T}^{-1} = \mathbf{T}^\dagger$, and knowledge of $\mathbf{u}^{gs}$ is equivalent to knowing one row of the TM[29].

Memory effects that laterally shift the output field (such as the tilt and shift memory effects) can be understood to arise from the existence of correlations in the real-space TM of the scatterer (or its Fourier transform), so that measuring one row of the TM also provides information about other rows[12]. Therefore once $\mathbf{u}^{gs}$ is found and a focus formed on the guide-star, these shift memory effects can be exploited to controllably scan the focus over a local area around the guide-star, known as the isoplanatic patch. The size of the isoplanatic patch is governed by the distance over which the output field can be translated before it de-correlates (i.e., the degree to which the approximation in Eq. (3) holds). This strategy has been used to create scanning imaging systems inside randomly scattering samples by a suitable combination of tilting and/or lateral translation of the input wavefront[30,31]. Therefore, in an initial answer to our second question (ii), so far it has been shown that diffraction limited imaging is possible using the shift–shift and tilt–tilt memory effects, which are capable of laterally translating the output field in two dimensions.

**Rotational and radial memory effects in multimode fibres**. We now consider the emergence of memory effects in other geometries, and examine an MMF as a key example. The near cylindrical symmetry of MMFs leads to the prediction of a basis in which the TM is approximately diagonal. For example, solving the wave equation in an idealised straight section of step index fibre at a single wavelength results in a set of orthogonal circularly polarised eigenmodes that maintain a constant spatial profile and polarisation on propagation[32]. Here, following work by Plöschner et al.[33], we refer to these modes as propagation invariant modes (PIMs). Even though real optical fibres differ from this idealised case, it was recently shown in ref. [33] that the TM of a short length of step index MMF is relatively diagonal when represented in the PIM basis.

Following our general framework, in this case $\mathbf{U} = \mathbf{P}$, the matrix transforming from the PIM basis to real-space, and so the

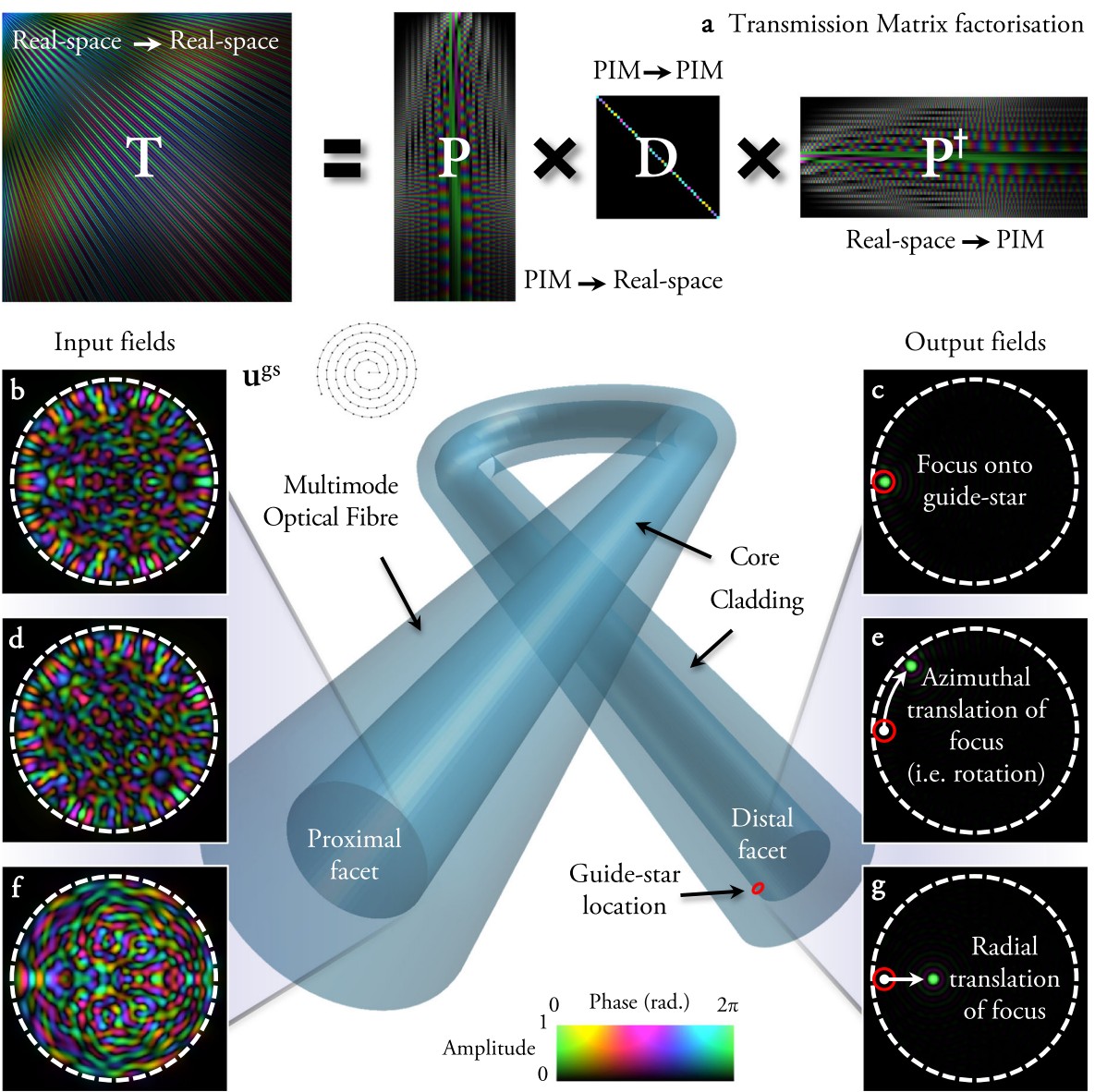

**Fig. 1 Field transformations through multimode fibres. a** The TM of an ideal MMF may be factorised into the product of three matrices. Shown is a low-dimensional example of these matrices for an MMF that supports 42 modes per polarisation at a wavelength of 633 nm. The scale-bar (shown at the bottom of the figure) indicates the amplitude (encoded in the brightness) and phase (encoded in the colour) of each complex matrix element. When real-space pixels are ordered outwards along an Archimedes spiral (example shown inset), translational correlations in the real-space TM are highlighted, i.e., a single row is similar to a spatially shifted copy of adjacent rows. **b** An example of a proximal field $\mathbf{u}^{gs}$ required to form a focus (shown in green in (**c**)) onto a guide-star at the edge of the distal facet of an ideal fibre (position of guide-star indicated by a red circle). The scale-bar representing the complex value of the electric field at the proximal and distal facets is the same as described in (**a**). Due to the rotational memory effect, by rotating the input field around the fibre axis (**d**) the output field is rotated by the same angle, moving the focus azimuthally (**e**). Using knowledge of $\mathbf{u}^{gs}$ and **P** enables estimation of the TM of the system, and prediction of the proximal field (**f**) required to radially translate the focus (**g**). These simulations model an ideal fibre supporting 754 modes per polarisation at a wavelength of 633 nm—this mode capacity is used for all simulations throughout the rest of the paper.

TM of an MMF can be factorised according to $\mathbf{T} = \mathbf{PDP}^\dagger$, where as before **D** is quasi-diagonal, and here we have used the conjugate transpose $\mathbf{P}^\dagger$ in place of the inverse $\mathbf{P}^{-1}$ under the assumption that **P** is sufficiently oversampled in real-space such that $\mathbf{P}^\dagger\mathbf{P} \propto \mathbf{I}$. (This is because the fibre modes are continuous functions of position, and orthonormal with respect to an integral over all space. Therefore as the sampling in real space is increased, the matrix multiplication $\mathbf{P}^\dagger\mathbf{P}$ becomes an increasingly better approximation to the set of mode overlap integrals, and therefore ever closer to the form of the identity matrix). Figure 1a shows a simulated low-dimensional example of these matrices for an MMF that supports 42 modes per polarisation at a wavelength

of 633 nm. Circularly polarised PIMs are specified by an azimuthal and radial index, $\ell$ and $p$ respectively. For a step index fibre of radius $a$, the transverse field of $\psi_{\ell,p}$, in cylindrical coordinates (denoted by radial position $r$ and azimuthal position $\theta$), is given by

$$\psi_{\ell,p}(r,\theta) = N_{\ell,p}e^{i\ell\theta}\begin{cases} \mathcal{J}_\ell(u_{\ell,p}r/a)/\mathcal{J}_\ell(u_{\ell,p}) & \text{for} \quad r < a \\ \mathcal{K}_\ell(\omega_{\ell,p}r/a)/\mathcal{K}_\ell(\omega_{\ell,p}) & \text{for} \quad r \geq a, \end{cases} \tag{4}$$

where $N_{\ell,p}$ is a normalisation constant ensuring each mode has a total intensity equal to 1, $\mathcal{J}_\ell$ is a Bessel function of the first kind of order $\ell$, and $\mathcal{K}_\ell$ is a modified Bessel function of the second kind

of order $\ell$. $u_{\ell,p}$ and $\omega_{\ell,p}$ are normalised transverse wave-numbers of each PIM in the core and cladding respectively, which may be found from the roots of the characteristic equation describing the parameters of the fibre (see ref. [32] and Methods for more details). $\ell$, the vortex charge, is proportional to the amount of orbital angular momentum (OAM) carried by the PIM, which is characterised by a helical wavefront that accrues a phase change of $2\pi\ell$ around one circuit of the fibre axis[34].

Equation (3) leads to an input modulation of $\mathbf{O} = \mathbf{PMP}^{\dagger}$, which tells us that phase changes imparted to individual PIMs at the input of the fibre are preserved at the output. Applying a phase ramp modulation linearly proportional to azimuthal mode index $\ell$, i.e., $\mathbf{M} = \text{diag}[e^{-i\boldsymbol{\ell}\delta\phi}]$, is equivalent to rotating the input field around the fibre axis by an angle $\delta\phi$. Here $\boldsymbol{\ell}$ is a vector specifying the vortex charge of each PIM. As these $\ell$-dependent phase changes are preserved through the fibre, the output field is also rotated by $\delta\phi$, as shown in Fig. 1b–e. This effect is already well-known, and coined the rotational memory effect[35]. Our general framework tells us that it is a rotational analogue of the shift-shift memory effect: the angular dependence of the MMF TM is quasi-diagonal in the 1D cylindrical Fourier basis (i.e., the OAM basis), and therefore an angular shift of the input field leads to an angular shift of the output field. Figure 1a shows the corresponding diagonal correlations in the real-space TM, when the real-space pixels are ordered along, for example, an Archimedes spiral, winding out from the central axis of the fibre (see spiral inset in Fig. 1).

We might wonder whether there is a corresponding effect where we can displace the output field from the fibre in a radial direction. Equation (3) highlights that the form of any memory effect based field transformation is intrinsically linked to the basis in which the TM is diagonal. As shown in all cases above, a displacement operation at the output is diagonalized in the Fourier basis. Therefore, without knowledge of the TM, any lateral displacement of the unknown output field is only possible in systems where the TM is quasi-diagonal in the Fourier basis. The radial dependence of the MMF TM is not quasi-diagonal in the Fourier basis. This means that a 'radially–shifting' memory effect, that serves to translate the entire output field of an MMF radially, does not occur—and in any case, such translations would be ill-defined at the fibre axis and core-cladding interface. In Supplementary Note 1, we explore the disruption of shifting memory effects in MMFs in more detail. We show how the presence of the fibre edges—the core-cladding interface—breaks the translational symmetry thus disrupting the equivalent of the 2D shift-shift memory effect in MMFs.

Despite this, modulations that operate on the field in the radial direction do exist. For example, consider the effect of a modulation based on the radial mode index $p$ on the input beam: $\mathbf{M} = \text{diag}[e^{i\mathbf{p}\delta\rho}]$, i.e., a phase ramp linearly proportional to $p$, where vector $\mathbf{p}$ holds the radial index of each PIM, and where $\delta\rho = \pi\delta r/a$, and $\delta r$ is radial distance across the fibre facet. Such a modulation may be viewed as a 'quasi-radial' memory effect: although as described above, it can't enable a simple translation of the distal field.

Figure 2 shows the behaviour of the quasi–radial memory effect. Intriguingly, if the output field is concentrated to a diffraction limited spot centred on the fibre axis, then the above modulation on the input field does indeed correspond to a radial shift of this point—i.e., it is transformed into a ring of intensity of radius $\delta r$ (see Fig. 2a). However, output fields corresponding to diffraction limited points at other radii on the distal facet show a variety of different transformations, depending on the radial coordinate of the point (see Fig. 2b–d). We emphasise that all of these output transforms can be achieved using the same input modulation, regardless of the length of the fibre. This tells us that

if the field is localised to a diffraction limited spot anywhere on the distal facet, the intensity pattern can be deterministically transformed—with a fidelity governed once again by how well the approximation in Eq. (3) holds.

In Supplementary Note 2 we investigate two other quasi-radial memory effects: Firstly we show that if the position of the initial focussed spot is known, then this information enables a phase-only modulation linearly proportional to the radial component of the wave-vectors composing the modes, $k_r$, to be applied - which is capable of moving the spot with less distortion than shown in Fig. 2 (although still not perfectly). Secondly we show that a modulation of the form $\mathbf{M} = \text{diag}[\cos(\mathbf{u}_{\ell,p}r/a)]$ enables a point to be radially expanded about its initial position, wherever that may be on the distal facet. Yet for an arbitrary speckled output beam, these radially varying transformations all interfere in a way that cannot be predicted without knowledge of the amplitudes and phases of the original distal speckle field - which itself requires knowledge of the entire TM.

**Guide-star assisted imaging through MMFs.** The lack of a true 'radially–shifting' memory effect has implications for imaging through uncalibrated MMFs. As scanning of unknown speckle patterns is only permitted in 1D (azimuthally), then there is no way to perform imaging based on 2D scanning of these speckle patterns as in ref. [20]. However, provided we know how to focus the field at the output to a point, the existence of the quasi-radial memory effects suggests a form of ghost imaging might be possible. For example, a sequence of known intensity patterns could be projected to the distal end of the fibre using the rotational and quasi-radial memory effects, and their level of overlap with the distal scene recorded back at the proximal end by measuring the total intensity of the return signal, allowing an image to be computationally reconstructed[36,37]. Although this idea holds promise, we now show that it is possible to do even better, and use the information held in matrix $\mathbf{P}$ (i.e., knowledge of the basis in which the TM is quasi-diagonal), to scan a well-defined focus around a guide-star in both azimuthal and radial directions. This opens up a 2D isoplanatic patch at the output of the fibre, within which scanning imaging is possible.

We imagine placing a single guide-star, such as a fluorescent particle, at the distal facet of the fibre (see the red circle on the fibre schematic in Fig. 1). The proximal field that will focus onto the guide-star, $\mathbf{u}^{gs}$, can be measured using phase stepping holography, with access only to the proximal end, as described in, for example, ref. [4], or in a single shot using phase conjugation[2].

So far, guided by Eq. (3), we have restricted ourselves to transforms on the input field of a particular form $\mathbf{O} = \mathbf{UMU}^{-1}$. We now relax this and consider more general transformations of the input field to achieve a radial shift in the position of a focussed spot on the distal facet. To accomplish output field modulations beyond what is possible with conventional memory effects alone (such as in this case), knowledge of the full TM of the scatterer is required - see Supplementary Note 1 for more discussion. Fortunately, it is indeed possible to obtain an estimate of the TM using the information at our disposal. In the case of an MMF, we can make the assumption that the TM is unitary (i.e., power loss is minimal). We also assume the real fibre's TM to be perfectly diagonal in the PIM basis. Under these assumptions, we can now find $\mathbf{D}'$, a diagonal approximation to the true quasi-diagonal TM $\mathbf{D}$. To proceed, we show that the measurements on the guide-star reveal the phase delay of each PIM due to propagation through the MMF. This phase information can then be used to populate the complex elements along the main diagonal of $\mathbf{D}'$, while setting all off-diagonal elements to zero.

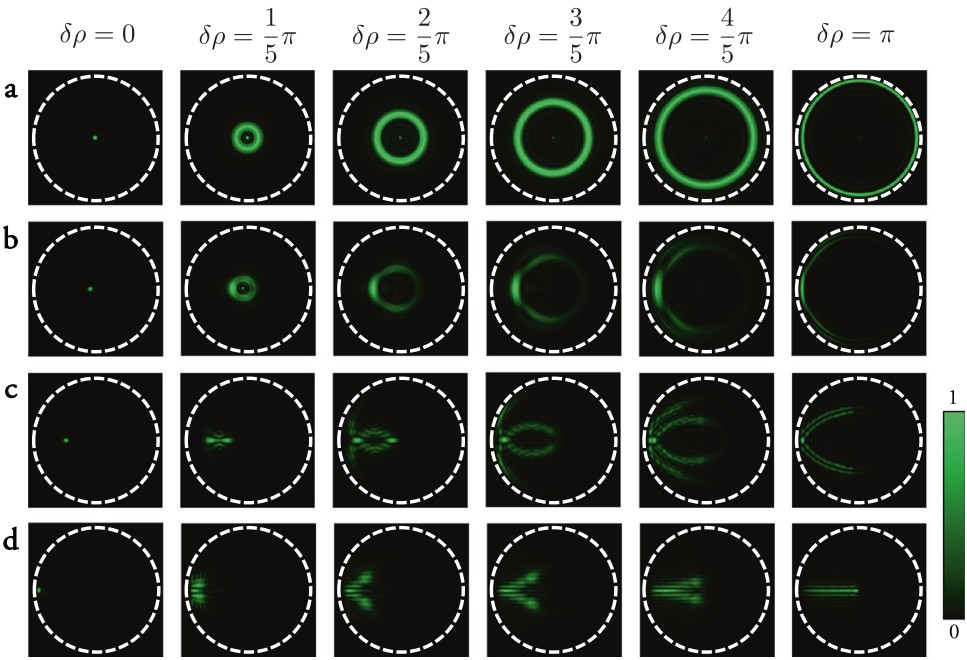

**Fig. 2 The quasi-radial memory effect in MMFs.** Each row shows simulations of a sequence of distal intensity patterns obtained by applying an input modulation $\mathbf{O} = \mathbf{PMP}^{-1}$ using $\mathbf{M} = \mathrm{diag}[e^{i\mathbf{p}\delta\rho}]$, for increasing $\delta\rho$, where the initial distal field (left most panel) is a diffraction limited spot at four different radial coordinates: on the fibre axis (**a**), just off fibre axis (**b**), midway between fibre axis and core-cladding boundary (**c**), and near core-cladding boundary (**d**). The scale-bar indicates the relative intensity within each panel. We see a variety of different predominantly radial transformations take place. We emphasise that all of these transformations are independent of the length of the fibre (i.e., independent of the complex values of the elements of $\mathbf{D}$). We also note that if the initial spot position is rotated about the fibre axis, so are these output fields, by virtue of the rotational memory effect[35].

More specifically, the defining equation for the input field $\mathbf{u}^{\mathrm{gs}}$ that focuses onto a guide-star is

$$\mathbf{Tu}^{\mathrm{gs}} = \mathbf{PDP}^{\dagger}\mathbf{u}^{\mathrm{gs}} = \mathbf{v}^{m_0}, \tag{5}$$

where we denote the output field focussed onto the guide-star as $\mathbf{v}^{m_0}$, which has all elements equal to zero except for one at the spatial point corresponding to the location of the guide-star, indexed by $m_0$. Substituting the perfectly diagonal approximation $\mathbf{D}'$ that we wish to find in place of the real quasi-diagonal TM $\mathbf{D}$, and multiplying both sides of Eq. (5) by the adjoint of $\mathbf{P}$ then gives $\mathbf{D}'\mathbf{P}^{\dagger}\mathbf{u}^{\mathrm{gs}} = \mathbf{P}^{\dagger}\mathbf{v}^{m_0}$. Our assumption of the unitarity of the fibre TM means that the elements of the diagonal matrix $\mathbf{D}'$ only vary by their phase term, i.e., the $n^{th}$ element $D'_{nn} = \exp(\mathrm{i}\phi_n)$. Therefore we can also re-write the defining equation for $\mathbf{u}^{\mathrm{gs}}$ as

$$e^{\mathrm{i}\phi_n}\sum_m P^{\dagger}_{nm}u^{\mathrm{gs}}_m = P^{\dagger}_{n,m_0}. \tag{6}$$

If we can estimate the PIM basis $\mathbf{P}$ and we know the input field $\mathbf{u}^{\mathrm{gs}}$ that focuses onto the guide-star at $m_0$, we can estimate the phases on the diagonal of $\mathbf{D}'$ by rearranging Eq. (6) for $e^{\mathrm{i}\phi_n}$, yielding

$$D'_{nn} = e^{\mathrm{i}\phi_n} = e^{\mathrm{i}(\sigma_n - \gamma_n)}, \tag{7}$$

where the two contributions to the phase are given by

$$\sigma_n = \arg\left[P^{\dagger}_{nm_0}\right], \tag{8}$$

$$\gamma_n = \arg\left[\sum_m P^{\dagger}_{nm}u^{\mathrm{gs}}_m\right]. \tag{9}$$

Here $\gamma_n$ is the phase delay mode $n$ picks up on propagation through the fibre, and $\sigma_n$ is a mode dependent phase correction term, equal to the relative phase required for each PIM to constructively interfere at the lateral location of the guide-star if it were positioned at the proximal end of the MMF. The intuition

behind $\sigma_n$ is to ensure that we incorporate only the phase delay due to the scatterer into the estimated TM, and remove any apparent phase delay associated purely due to the choice of diagonal basis. The *Approximate Transmission Matrix* (ATM) of the MMF in the real-space basis is then given by $\mathbf{T}' = \mathbf{PD}'\mathbf{P}^{\dagger}$. The ATM can now be used to predict the proximal field modulations required to move the focus both azimuthally and radially (see Fig. 1f, g). Evidently, as will be shown later, knowledge of the ATM gives more general control over the output field than simply shifting the position of a focus.

The fidelity of the ATM will clearly depend upon how well we can estimate the basis in which the TM is diagonal (i.e., the validity of the approximation in Eq. (3)). Figure 3 shows simulations of the imaging performance through an MMF as our ability to accurately estimate the diagonal basis is compromised. We simulate the measurement of $\mathbf{u}^{\mathrm{gs}}$ using a guide-star placed at the core-cladding boundary (marked by a red circle in Fig. 3), and construct the ATM. See Methods for a detailed description of these simulations, which capture how power is spread away from the diagonal of the TM in a realistic manner. Figure 3a–f shows the power-ratio $p_{\mathrm{r}}$ - the ratio of power focussed into a target spot using the ATM compared to total power transmitted to the distal facet, as $p_{\mathrm{d}}$, the percentage of power on the main diagonal of the real TM $\mathbf{D}$, decreases. Each panel maps how the power-ratio varies over the distal facet - capturing the size and shape of the isoplanatic patch. We see that the error in the ATM $\mathbf{T}'$ is non-uniformly distributed over its columns— meaning that columns representing points close to the guide-star are well captured, while the errors in columns representing points further away grow with distance from the guide-star. This results in an isoplanatic patch that gradually collapses around the location of the guide-star as $p_{\mathrm{d}}$ reduces. Figure 3g–l shows simulations of the guide-star based imaging performance, indicative of the spatial variation in contrast and resolution in

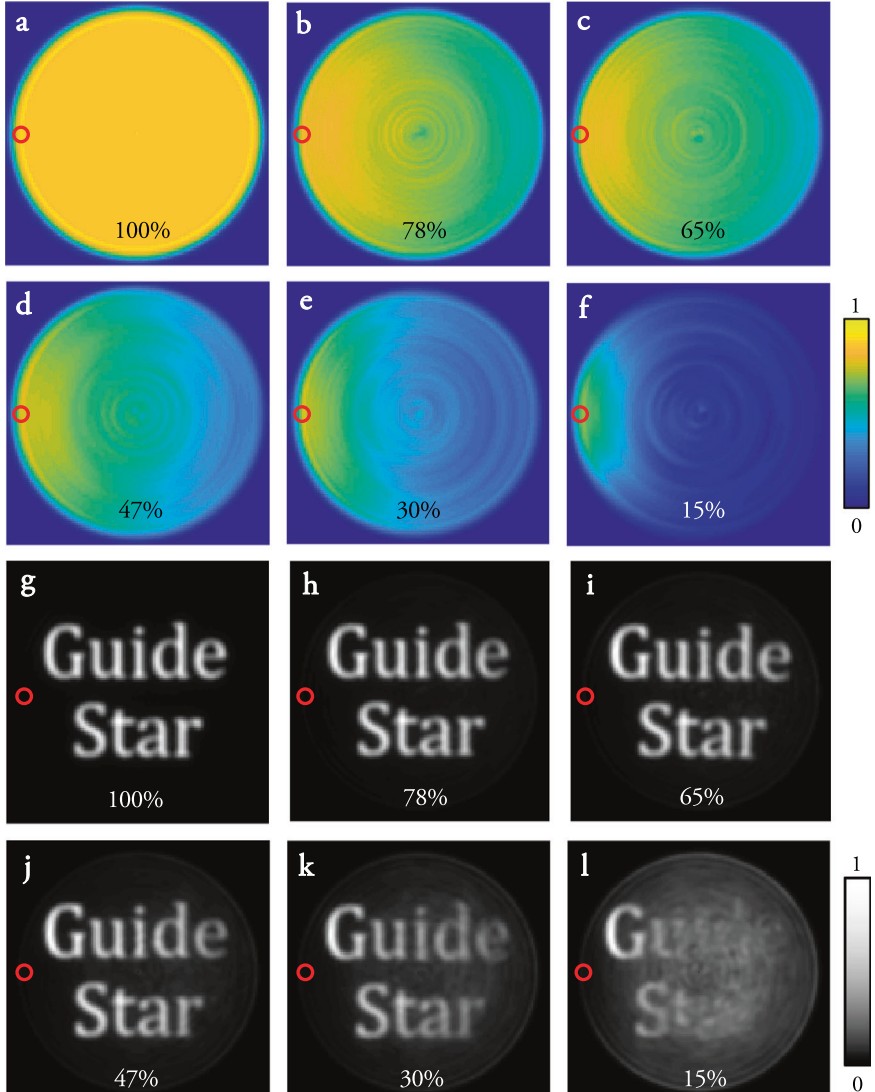

**Fig. 3 The size and shape of the isoplanatic patch:. a–f** Simulated maps of the power-ratio of points focussed to different regions of the distal facet through non-ideal fibres with quasi-diagonal TM **D**. The scale-bar indicates the power-ratio across the distal facet. Higher values of the power-ratio indicate regions on the distal fibre facet where spot scanning can be achieved with greater contrast, leading to higher fidelity imaging. We see that the isoplanatic patch gradually shrinks around the location of the guide-star (marked by a red circle). Foci are created using ATM **D′** calculated from **u**$^{\text{gs}}$ using Eq. (7). $p_{\text{d}}$, the power on the diagonal of **D**, decreases from (**a**) to (**f**) and is given at bottom of each panel. **g–l** Simulations of scanning imaging capabilities in each case. The scale-bar indicates the relative intensity of the reconstructed images.

each case. Here imaging across the entire facet appears to be disrupted once $p_{\text{d}}$ falls below ~25%.

We now consider the factors that govern the area of the isoplanatic patch at the distal facet of the MMF. The angular extent of the tilt–tilt memory effect, $\alpha$, is related to a single parameter describing the sample: the thickness of the scattering layer $L_{\text{s}}$, so that $\alpha \sim \lambda/(2\pi L_{\text{s}})$, where $\lambda$ is the incident wavelength[13]. This raises the question of whether it is also possible to derive a similar equation describing the limits of the memory effects in MMFs. We start by noting that, as shown in Fig. 3a, an ideal perfectly straight MMF supports a rotational memory effect of $2\pi$, and a quasi-radial memory effect of $r$, regardless of its numerical aperture, core diameter or length. However, the size and shape of the isoplanatic patch exhibited by real MMFs is highly sensitive to a large number of parameters. These include misalignments of the optical system at the input and output of the fibre (at least 6-degrees of freedom at each end), the departure of the cross-sectional refractive index profile of the

fibre from the ideal (or assumed) case, how this profile varies along the fibre's length, and the bend configuration of the fibre[33]. This renders finding an analytical expression for the extent of the memory effect in MMFs a challenging task. Nevertheless, the combined effect of these parameters is to spread power into the off-diagonal elements of the real TM **D**, and we find that the approximate area of the isoplanatic patch, $A_{\text{iso}}$, is related to the fraction of power remaining on the diagonal of the actual TM, $p_{\text{d}}$, according to: $A_{\text{iso}} \sim \pi r^2 p_{\text{d}}$. See Methods for a derivation of this expression and the approximations used. This trend is qualitatively apparent in Fig. 3.

**Proof of principle experiments**. In order to test our concept in practice, we design an experiment capable of measuring the TM of a step index MMF at a single circular input and output polarisation. We study an MMF of NA = 0.22, core diameter = 50 $\mu$m, and $L$ ~ 30 cm in length, supporting 754 spatial modes per

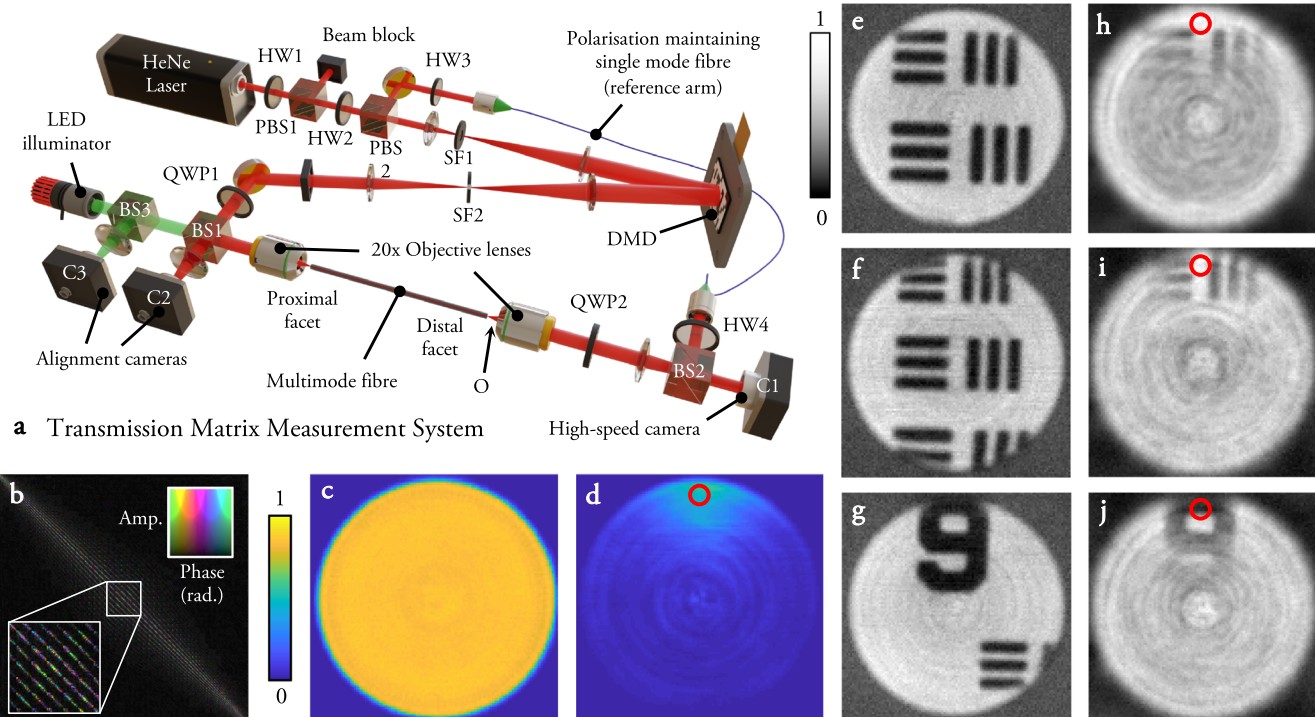

**Fig. 4 Proof of principle experiments. a** Experimental set-up to measure the full TM of an MMF, emulate ATM measurement with a guide-star, and image through the MMF using the full TM and ATM. HW half wave-plate, QW quarter wave-plate, BS beamsplitter, PBS polarising beamsplitter, SF spatial filter. Supplementary Note 3 provides a more detailed description of the set-up. **b** Fully sampled TM of an MMF represented in PIM basis. The scale-bar is the same as that used in Fig. 1, and here indicates the amplitude (encoded in brightness) and phase (encoded in colour) of each matrix element. Inset shows an enlarged region of the TM indicating the fine structure around the main diagonal. Supplementary Fig. 4 shows an enlarged plot of this TM. Supplementary Movie 1 shows the input and output fields used to fully sample the TM, which also visually highlights the symmetries present in the MMF transform. Power ratio maps of scanned foci using the fully sampled TM (**c**) and guide-star based ATM (**d**). The scale-bar indicates the power-ratio across the distal facet. Guide-star location is marked with a red circle. **e–g** Example transmission images of a resolution target obtained with the fully sampled TM. The location where the resolution target is inserted at the distal facet of the fibre is marked with an O. **h–j** Examples of transmission images of the same scene as (**e–g**), in this case obtained with guide-star based ATM. The scale-bar indicates the relative intensity of the reconstructed images. Supplementary Movie 2 shows the focus being scanned in 2D across the isoplanatic patch at the distal facet of the MMF.

polarisation at a wavelength of 633 nm. These dimensions are chosen as a typical example of an MMF that may find micro-endoscopic imaging applications. Since the input circular polarisation is maintained over propagation through short lengths of fibre, we sample an orthogonal sub-space of the full TM (i.e., a single polarisation), which is sufficient for high-contrast imaging behind the distal end[33]. The experimental set-up is similar to that described in ref. [38] and is shown in Fig. 4a. The spatial mode of input light is controlled using a digital micro-mirror device (DMD) in the Fourier plane of the proximal fibre facet, and the distal fibre facet is imaged onto a high-speed camera, along with a coherent reference beam enabling extraction of the complex field via phase stepping holography (See Methods and Supplementary Notes 3 and 4 for more details. Supplementary Note 9 details the loss throughout the experimental system).

We first fully sample the TM of the MMF at a single polarisation. Supplementary Movie 1 shows the input probe fields, and holographically reconstructed output fields during this measurement. After correcting for coarse misalignments in the input and output beams following the strategy outlined in ref. [33] (we note this does not require an optimisation procedure), and transforming to the PIM basis representation, the fully sampled TM possesses $p_d \sim 15\%$ of its power on the main diagonal (TM shown in Fig. 4b). The degree of mode coupling in the TM can also be quantified by the ratio of $L/\ell_f$, where $\ell_f$ is the TMFP in the fibre mode (PIM) basis, i.e., the estimated length of fibre beyond which the TM can be considered fully coupled. In this case we

find our experimentally measured TM has a level of mode coupling consistent with $L/\ell_f \sim 0.02$, a value we estimate by following the methods given in ref. [39]. Therefore, in this experiment we anticipate a guide-star assisted imaging performance equivalent to that predicted in Fig. 3f—i.e., an isoplanatic patch extending around the guide-star, but not reaching across the entire output facet of the fibre.

Figure 4c and d shows the experimentally measured power ratio maps of foci generated at the distal facet of the fibre using the fully sampled TM 4(c), and a guide-star based ATM (Fig. 4d). Supplementary Movie 2 shows the focus as it is scanned in 2D across the isoplanatic patch at the distal facet of the MMF. The ATM was constructed from data recorded by a single pixel of the camera imaging the output facet, mimicking the action of a guide-star. The extent of the isoplanatic patch in Fig. 4d matches well with Fig. 3f as predicted. Figure 4e–g shows transmission images of a resolution target using the fully sampled TM. Figure 4h–j shows the same scenes imaged using the ATM (more detail of how these images was recorded is given in Methods). As expected, we see that imaging is possible through a small isoplanatic patch around the guide-star, and objects outside this patch are not discernible as we are unable to create a high contrast focus in these regions. The gradual reduction in resolution and contrast away from the guide-star is analogous to the situation found in memory effect based imaging through thin scattering layers.

We note that in these experiments, the object (resolution target) is positioned ~40 $\mu$m away from the distal facet of the fibre. Therefore, in order to reconstruct an image of the object in focus, the scanned foci had to be axially refocussed from the distal facet to the object plane. Due to the cylindrical symmetry of the fibre, the radial k-vector $\left(k_r = \sqrt{k_x^2 + k_y^2}\right)$ of input light is approximately preserved at the output—which is a consequence of the quasi-diagonal nature of the fibre TM in the PIM basis. As previously demonstrated in refs. [17,40], refocussing can be achieved by taking advantage of this symmetry and adding a quadratic 'lensing' phase term directly to the field that is generated by the DMD. We note this input transformation is also related to the recently reported 'chromato-axial' memory effect studied in other forward scattering media[41]. Therefore we actually create fields corresponding to defocussed spots at the distal facet of the fibre, which then focus onto the resolution target. Despite this increase in complexity, Fig. 4h–j demonstrates that this refocussing can be successfully achieved using the ATM, which affords a high level of control over the field within the isoplanatic patch. Supplementary Note 5 discusses the refocussing capability in more detail.

In addition to imaging, it is also possible to project patterns through the MMF into the isoplanatic patch, examples of which are described in Supplementary Note 6. We also studied the effect of fibre deformation on guide-star assisted focusing, and found only a minor reduction in the power-ratio of foci was observed when the fibre was bent through 90° with a radius of curvature of ~10 cm, as described in Supplementary Note 7. This result shows there is potential for applying single-ended guide-star based TM calibration to image through flexible MMFs undergoing time varying deformations. More generally, these refocussing and pattern projection results show that point-spread-function engineering should be possible within the isoplanatic patch[42].

The location of the guide-star on the distal facet also plays a key role in determining the accuracy of the ATM, and thus the size and shape of the isoplanatic patch. Each focus is formed as a superposition of PIMs which interfere constructively. We can only measure the phase change of PIMs that have an intensity profile that overlaps with the guide-star position. Therefore, it is preferable to choose a guide-star location that overlaps with the highest number of PIMs. Inspecting the rows of matrix **P**, which hold this overlap information, reveals that guide-star locations at the core-cladding boundary overlap with all of the PIMs - although we note that some PIMs of index $\ell = 0$ will have very low intensity at the edge of the core as their fields are concentrated mainly on the fibre axis. Therefore we have chosen a guide-star positioned at the core-cladding boundary.

It is straight forward to test alternative guide-star positions in our proof-of-principle experiments—moving the guide-star simply means choosing a different camera pixel to use for ATM construction. Figure 5 shows the experimental power ratio maps for three different choices of guide-star location, along with the theoretical level of overlap with the PIMs in each case. As expected, the area of the isoplanatic patch is maximised when the guide-star is placed on the core-cladding boundary (Fig. 5a), enabling recovery of most of the diagonal elements of the ATM in the PIM basis. As the guide-star location is moved radially inwards towards the centre of the core, the number of PIMs sampled by the guide-star progressively decreases (Fig. 5a–c). When the guide-star is placed at a radius of $a/2$, we observe a strong arc in the isoplanatic patch, a signature of the rotational memory effect (Fig. 5e). This isoplanatic arc is formed as projection of foci to the same radius as the guide-star require the same combination of PIMs, the phase delays of which have been accurately sampled in the ATM. However we also observe a sharp reduction in the power-ratio when the focus is moved radially in

Fig. 5e as these points require the constructive interference of different combinations of PIMs, some of which are not well-sampled by the guide-star. When the guide-star is located on the fibre axis, it exclusively samples only the PIMs with a vortex charge of $\ell = 0$ (Fig. 5c), and the isoplanatic patch contracts to a single spot as shown in Fig. 5f.

There is one point on the distal facet that we know how to focus on perfectly: the position of the guide-star itself, which has been directly measured. However, as can be seen in both simulations and experiments in Figs. 3–5, as $p_d$ decreases, then using the ATM does not yield a perfect focus even onto the guide-star itself. This is a consequence of our assumptions that the fibre TM is perfectly diagonal and unitary - assumptions that become less accurate with lower values of $p_d$. To mitigate this, when scanning over the guide-star location, the proximal field returned by the ATM can be replaced with $\mathbf{u}^{gs}$. Points at the same radius as the guide-star may also be created with a higher power ratio than achievable with the ATM, by rotating the field used to generate the guide-star focus around the fibre axis - i.e., directly employing the rotational memory effect[35]. This observation also points to a more sophisticated approach to reconstruct the ATM: instead of forcing $\mathbf{D}'$ to be diagonal, we could try to iteratively search for a non-diagonal and potentially non-unitary $\mathbf{D}'$, allowing the minimum power in off-diagonal elements[43], so that $\mathbf{PD}'\mathbf{P}^\dagger$ also perfectly focusses onto the guide-star when operating on input field $\mathbf{u}^{gs}$. This is a severely under-constrained problem that we plan to investigate in more detail in the future.

Equation (7) stipulates that we must know the location of the guide-star on the distal facet in order to apply the phase correction term $\boldsymbol{\sigma}$. However, if the location of the guide-star is unknown and $\boldsymbol{\sigma}$ is omitted (i.e., $\boldsymbol{\sigma} = 0$), $\mathbf{D}'$ still provides an estimate of the TM of the MMF, but with an unknown rotation between the proximal and distal planes. We note that in this case, the approximate radial location of a guide-star at an unknown position can be retrieved, as it is uniquely encoded in the relative amplitudes of the PIMs overlapping with the guide-star, which is extracted by decomposing $\mathbf{u}^{gs}$ in the PIM basis. This implies that the methods presented here may be possible even without specifically engineering a guide-star on the distal facet of the fibre. For example, a guide-star may be formed at a random location by optimising a non-linear feedback signal at the proximal end of the fibre[44]. Once the proximal field required to generate a focus, $\mathbf{u}^{gs}$, is found, the focus can then be scanned in 2D using the methods we have presented here.

## Discussion

In summary, we have demonstrated the possibility of guide-star assisted imaging through multimode fibres, which can be calibrated with access only to the proximal end of the fibre. We believe this technique to hold promise for the development of flexible multimode fibre based imaging systems. The reconstruction of the ATM, and of the images through the fibre, requires no iterative phase retrieval, and invokes no assumptions about the statistical properties of the scene at the end of the fibre —instead relying on assumptions about the properties of the fibre itself. Using a high-speed DMD capable of modulating input light at ~ 20 kHz, measurement of the proximal field that focuses on a fluorescent guide-star, $\mathbf{u}^{gs}$ (from which the ATM is calculated) consisting of, for example, ~1000 modes can be performed in ~200 ms using phase stepping holography[38,45]. Using a highly reflective guide-star, the same calibration could potentially be achieved in a single shot—albeit with a lower SNR[2,23] (Supplementary Note 8 contrasts these two methods). Therefore there is potential for both calibration and imaging to be performed in real-time.

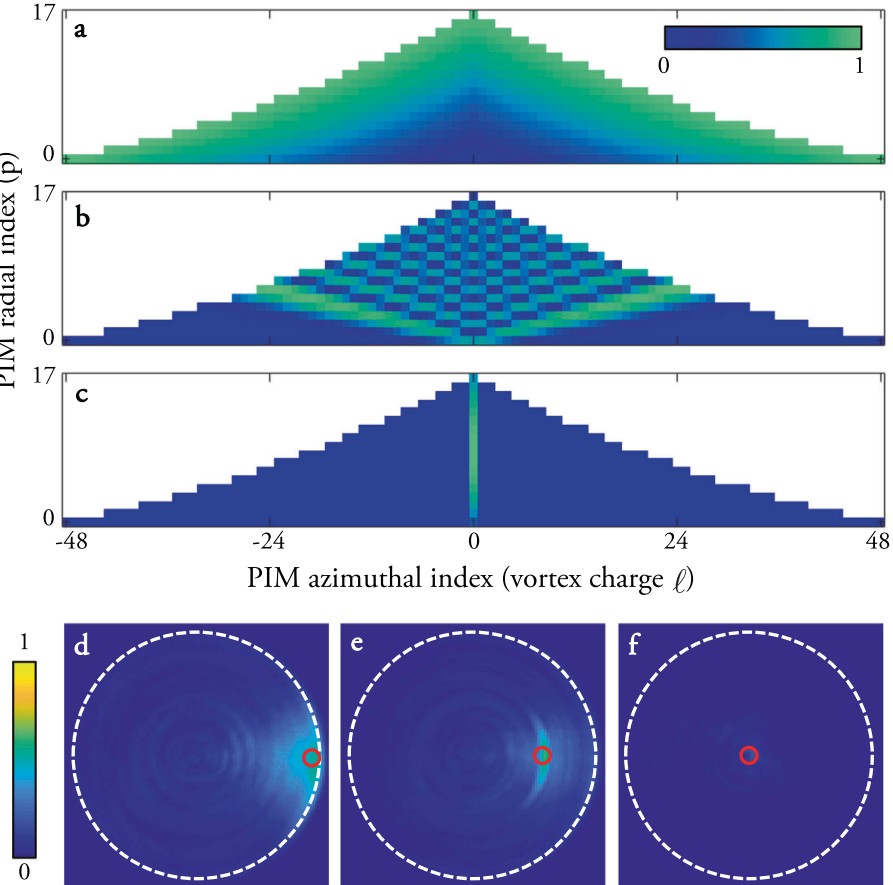

**Fig. 5 Choice of guide-star location. a–c** Theoretical absolute values of the level of overlap of all PIMs with three different guide-star locations. The scale-bar indicates the relative amplitude of the each PIM. Maps (**a–c**) corresponds to guide-star locations shown by red circles in (**d-f**). The colour of point $\ell, p$ in PIM maps represents the relative level of overlap of $\psi_{\ell,p}$ with the specified guide-star location. **d-f** Experimentally measured power-ratio maps using ATMs reconstructed from guide-stars in the three different locations, marked with red circles. The scale-bar indicates the power-ratio across the distal facet. A guide-star positioned at the core-cladding interface gives the greatest level of overlap with all of the PIMs and therefore the largest isoplanatic patch. A guide-star positioned on the fibre axis achieves the lowest level of overlap with the PIMs, and the isoplanatic patch shrinks to a single diffraction limited point in this case.

In our proof of principle experiments, the isoplanatic patch does not extend over the full area of the distal facet. However our simulations show that the area of this isoplanatic patch can be significantly increased by making a better estimate of the basis in which the TM of the MMF is diagonal (see Fig. 3). Such an estimate can be achieved by optimising the fibre parameters with access to the full TM, before subsequent deployment of the system as a flexible micro-endoscope[33,46], which will be a focus of our efforts in future work. To image through longer MMFs which possess higher levels of mode coupling (and therefore lower values of $p_d$ even after fibre parameter optimisation), we envisage that multiple guide-stars, distinguished by, for example, emission frequency or speckle pattern contrast[47,48], could be used to increase the total area of the distal facet through which it is possible to image. If necessary, accurate guide-star placement could potentially be achieved using, for example, laser-ablation-assisted attachment of fluorescent sensors[49].

We highlight that recently there have been several innovative proposals to achieve single ended TM characterisation of an MMF. For example, Gu et al. proposed recovery of the forward TM by measurement of the round trip reflection matrix of a fibre in which the distal facet has been engineered with a partial reflector of spatially varying reflection coefficient, and moveable shutter[50]. The same group more recently suggested a simpler approach with a uniform partial reflector that can go some way to improving spot

formation[51]. Gordon et al. proposed the placement of a stack of patterned spectral filters at the distal facet and recovering the forward TM by measuring the reflection matrix at multiple frequencies[52]. Chen et al. proposed placing a compact spatial multiplexer at the distal facet capable of imprinting a mode dependent time delay or frequency modulation[53]. The engineering requirements of these proposals are technically challenging, and so they have yet to be experimentally realised in MMFs of high enough resolution to be useful for imaging. The re-imaging properties of graded index fibres have also been demonstrated to enable a spot to be focussed at a chosen location within up to half of a distal facet using a non-linear feedback signal from two-photon fluorescence[54]. However, in this case, an optimisation process taking on the order of several minutes was required to achieve a focus at each new location. The guide-star assisted imaging concept we present here is arguably simpler than these recent proposals, although it may be more limited in terms of the accuracy of the forward TM that can be recovered, and the length of the fibre that may be calibrated. In addition to spatial correlations in weakly coupled MMFs, we also note that spatio-temporal correlations have been identified in MMFs with strong mode coupling that may prove useful for pulse delivery[55,56]. Ultimately, we hope that some combination of our imaging strategy, in conjunction with the alternative concepts above, may provide the most robust method to enable imaging through flexible MMFs.

Beyond applications to step index optical fibres, and in answer to our questions posed earlier, we have shown that guide-star assisted imaging need not invoke tilt or shift memory effects, and can be extended to any scattering system in which we have an estimate of the basis in which the TM is quasi-diagonal. Prior knowledge of this sort can be understood as a generalisation of the concept of the TMFP to spaces other than the momentum basis[39]. i.e., advance knowledge that a scattering system possesses an optical path length that is much lower than a 'generalised' TMFP in a known (arbitrary) basis. The diagonal nature of the TM indicates that spatial modes input in the diagonal basis are not randomised into the entire vector space describing the allowed modes within the structure. Our work provides a guide on how to efficiently hunt for and make use of memory effects in optical systems of arbitrary geometry. This concept may be applied to a range of other media including graded-index[57,58] and photonic crystal fibres[59], few-mode fibre bundles (which have the constraint that the guide-star is ideally placed in the far-field of the output[60]), photonic lanterns[61], opaque walls (i.e., imaging around corners)[62], and artificially engineered photonic networks and scattering systems.

## Methods

### Construction of matrix P.
To perform guide-star assisted imaging through MMFs, we must estimate matrix **P**, that transforms from the propagation invariant mode (PIM) basis to real-space. The PIMs are found by solving the wave equation in a cylindrical geometry. This reduces to finding the roots of the scalar characteristic equations describing the modal dispersion[32].

$$u \frac{\mathcal{J}_{\ell-1}(u)}{\mathcal{J}_\ell(u)} + \omega \frac{\mathcal{K}_{\ell-1}(\omega)}{\mathcal{K}_\ell(\omega)} = 0, \quad (10)$$

where $\mathcal{J}_\ell$ is a Bessel function of the first kind of order $\ell$, and $\mathcal{K}_\ell$ is a modified Bessel function of the second kind of order $\ell$. $u$ and $\omega$ are normalised transverse wave-numbers which are related to the fibre parameters through

$$u = a\left(k^2 n_{core}^2 - \beta^2\right)^{\frac{1}{2}}, \quad (11)$$

$$\omega = a\left(\beta^2 - k^2 n_{core}^2\right)^{\frac{1}{2}}, \quad (12)$$

$$\omega^2 = v^2 - u^2, \quad (13)$$

$$v = ak\,\text{NA}, \quad (14)$$

$$\text{NA}^2 = n_{core}^2 - n_{clad}^2. \quad (15)$$

Here $a$ is the radius of the core, NA is the numerical aperture of the fibre and $n_{core}$ and $n_{clad}$ are the refractive indices of the core and cladding respectively. In this work we use the manufacturer's values of $a = 25\,\mu m$ and NA=0.22. $k$ is the vacuum wavenumber: $k = 2\pi/\lambda$, where $\lambda$ is the vacuum wavelength. $\beta$ is the propagation constant describing the phase velocity of each mode. Equation (10) may have multiple roots $u_{\ell,p}$, indexed by the vortex charge $\ell$ specified in the order of the Bessel functions, and radial index $p$ which counts the roots for a particular choice of $\ell$. Once the roots have been found, the function $\psi_{\ell,p}(r, \theta)$ represents the complex 2D field profile of the mode indexed by $\ell, p$ according to Equation (4). Here $\omega_{\ell,p}$ are related to $u_{\ell,p}$ through Equation (13), and $r$ and $\theta$ are the radial and azimuthal coordinates respectively across the core. Roots with a real phase velocity $\beta_{\ell,p}$ are propagating modes within the core. $\mathcal{K}_\ell$ takes argument $\omega_{\ell,p}$, which is imaginary when $\beta_{\ell,p} < k^2 n_{core}^2$. This then describes the evanescent field in the cladding of the fibre (i.e., for the region $r \geq a$). The PIMs, indexed by $\ell, p$ are ordered into a 1D list, indexed by $n$. Once the 2D complex functions describing each PIM have been found, column $n$ of matrix **P** is constructed by representing the complex field of the $n^{th}$ PIM in Cartesian coordinates on a 2D grid, and reshaping it into a column vector.

To account for some of the experimental misalignments, misalignment operators are measured and applied at each end of the fibre that remove course position and tilt misalignments of the input and output. These operators are found following the coarse alignment methods in ref. [33]. In brief, we make use of the fact that both ends of the MMF are accessible before it is employed as an endoscope, and measure the full real-space TM of the fibre (at a single polarisation). This real space TM can be processed without optimisation to reveal estimates of the position and tilt misalignments of the inputs and outputs in the following ways: (i) The central position of the core at output can be found by measuring the centre-of-mass of the sum of the intensity of all output measurements. (ii) The tilt of the output can be found by measuring the centre-of-mass of the sum of the intensity of the Fourier transform of all output measurements. Both of these methods can be

understood as operations involving summing over the absolute square of the columns of the measured TM. Under the assumption that the TM is unitary, the transpose of this TM is equivalent to the TM if it were measured in the opposite direction through the fibre. Therefore the position of the core at the input, and the tilt of the input light can be found by performing the equivalent to (i) and (ii), but summing over the rows rather than the columns of the TM. Once these misalignments are found, they can be represented as misalignment matrices at the proximal end ($\mathbf{R}_{pr}$) and at the distal end ($\mathbf{R}_{dl}$), as described in more detail in Supplementary Note 4.

### Simulations.
The optical transformation properties of fibres in real-space are modelled following the methods in ref. [33], by decomposing real space input fields into the PIMs, multiplying by a TM represented in the PIM ↦ PIM basis (**D**), and transforming back to real space (see Fig. 1a): $\mathbf{T} = \mathbf{PDP}^\dagger$.

An ideal fibre (as simulated in Figs. 1 and 2), of length $L$ is modelled by a diagonal matrix **D**, where the $n^{th}$ diagonal element is given by $e^{iL\beta_n}$. To model non-ideal fibres with quasi-diagonal TMs in the PIM basis, as shown in Fig. 3, simulated misalignment matrices **R** are introduced that transform the real-space field at each end of the fibre into a new real-space basis that is laterally ($dx$ and $dy$) and axially ($dz$) shifted, and tilted (about $x$ and $y$), and defocussed a small randomly chosen amount. A quasi-diagonal $\mathbf{D}_q$ with realistic off-diagonal coupling is created by absorbing these misalignment matrices into the TM in the PIM ↦ PIM basis, i.e., $\mathbf{D}_q = \mathbf{R}_{dl}'^\dagger \mathbf{DR}_{pr}'$, where $\mathbf{R}_{pr}' = \mathbf{P}^\dagger \mathbf{R}_{pr}\mathbf{P}$ and $\mathbf{R}_{dl}' = \mathbf{P}^\dagger \mathbf{R}_{dl}\mathbf{P}$ are the two different misalignment operators at the proximal and distal end of the fibre respectively here represented in the PIM basis. Therefore to reduce the power on the diagonal ($p_d$) of $\mathbf{D}_q$, and spread it off the diagonal in a realistic manner, the magnitudes of the randomly chosen misalignment in each dimension are increased.

Figure 3 shows modelling of the reconstruction of the ATM, and the performance of using it to focus and image through the fibre $p_d$ is decreased. To simulate this, we use $\mathbf{T} = \mathbf{PD}_q\mathbf{P}^\dagger$ to calculate the field required to focus on the guide-star, $\mathbf{u}^{gs}$, where $\mathbf{u}^{gs} = \mathbf{T}^\dagger \mathbf{v}^{m_0}$, and column vector $\mathbf{v}^{m_0}$ represents the desired field in vectorised form (i.e., zeros everywhere except for 1 at the element $m_0$ corresponding to the location of the guide-star). $\mathbf{u}^{gs}$ is then used to reconstruct the ATM $\mathbf{T}'$ according to Eqs. (7)–(9). The ATM $\mathbf{T}'$ is then used to calculate the estimated input fields required to raster scan a focus across the distal facet of the fibre, which are then propagated through the real TM $\mathbf{T}$, enabling the power ratio maps and reconstructed images to be modelled.

### Binary amplitude hologram design.
A DMD is used to shape the light into the MMF during TM measurement and for imaging. To generate an arbitrary (bandwidth limited) spatially varying complex field $\mathcal{A} = A(x, y)e^{i\alpha(x,y)}$ in the first diffraction order in the Fourier plane of the DMD ($x$ and $y$ being Cartesian coordinates of real-space in the optical Fourier plane of the DMD), we encode the inverse Fourier transform of $\mathcal{A}$, i.e., $\mathcal{B} = \mathcal{F}^{-1}(\mathcal{A}) = B(x', y')e^{i\gamma(x', y')}$, into a binary amplitude hologram to be displayed on the DMD, $\mathbf{H}(x', y')$. Here $x'$ and $y'$ are Cartesian coordinates on the DMD chip. Following[63,64], $\mathbf{H}(x', y')$ is given by

$$\mathbf{H}(x', y') = \frac{1}{2} + \frac{1}{2}\,\text{sgn}\left[\cos(p(x', y')) - \cos(q(x', y'))\right], \quad (16)$$

where

$$p(x', y') = \gamma(x', y') + \phi_{tilt}(x', y'), \quad (17)$$

$$q(x', y') = \arcsin\left[\frac{B(x', y')}{B_{max}}\right], \quad (18)$$

and where $B_{max}$ is the maximum amplitude of $B$, and $\phi_{tilt}(x', y') = k_x x' + k_y y'$ is a phase gradient directing the desired beam into the first diffraction order at an angle defined by wave-vectors $k_x$ and $k_y$. This hologram **H** can be intuitively understood as one in which the local phase of the grating (i.e., local lateral position of the grating cycle) varies in proportion with the desired phase of the target complex field, and the local duty cycle of the grating varies as a function of the desired amplitude. **H** also results in light transmitted into other diffraction orders, which can be spatially filtered in the Fourier plane of the DMD, leaving only the desired optical field transmitted into the rest of the optical system. The wave-vectors $k_x$ and $k_y$ also place a bandwidth limit on the complex field that can be generated and still successfully separated from other diffraction orders in the Fourier plane of the DMD.

### Imaging through guide-star calibrated MMFs.
The experimental setup, along with TM and ATM measurement are described in detail in Supplementary Notes 3 and 4. Once the TM and ATM are measured, they are used to capture the scanning imaging results presented in Fig. 4. A transmissive resolution target mounted on a manually operated 3D translation stage was manoeuvred to ~ 40 μm from the distal facet of the fibre. The scanning spot was refocused from the distal fibre facet to the plane of the resolution target by adding a quadratic Fresnel lens phase function to the hologram displayed on the DMD. The fully sampled TM or the ATM was used to calculate the proximal fields required to attempt to raster scan the focus across the resolution target. At each spot position $x, y$, the total

transmitted intensity arriving at camera C1 was recorded ($I(x, y)$), which directly represented pixel $x, y$ of the reconstructed image. We also attempted reflection based imaging by using a reflective resolution target and recording the total intensity of light reflected back through the fibre to camera C3. However we found that reflection images were very noisy due to the low laser power (1 mW) of the laser available for the experiment. Reflection imaging through MMFs has been successfully demonstrated in the past, but due to the low numerical aperture of MMFs, signals are typically very low when imaging beyond the end of the fibre facet, as we were doing here, and so laser powers on the order of at least several hundred mW are necessary for the return signal to be accurately measured. Reflection imaging using the ATM should be possible in future work using higher power lasers.

**The extent of the isoplanatic patch in MMFs**. The approximate area of the isoplanatic patch, $A_{iso}$, is related to the fraction of power remaining on the diagonal of the actual TM, $p_d$, according to: $A_{iso} \sim \pi r^2 p_d$. This can be shown as follows:

$$p_d = \tfrac{1}{N_1} \text{Tr}[\mathbf{D}^\dagger \mathbf{D}] \sim \tfrac{1}{N_2} \text{Tr}[\mathbf{S}^\dagger \mathbf{S}]$$
$$\sim \tfrac{1}{\pi a^2} \int_0^a \int_0^{2\pi} p_r(r, \theta) r \, d\theta dr \sim \tfrac{A_{iso}}{\pi a^2}. \tag{19}$$

Here $\text{Tr}[\cdot]$ refers to the trace of a matrix, and $N_1$ and $N_2$ are normalisation constants given by $N_1 = \sum_n \sum_m |D_{nm}|^2$ and $N_2 = \sum_n \sum_m |S_{nm}|^2$. We have introduced matrix $\mathbf{S} = \mathbf{L} \mathbf{T} \mathbf{T}^\dagger$, i.e., the real TM of the fibre represented with a change of input and output basis. The basis of $\mathbf{S}$ is chosen such that the input modes are the set of input fields predicted by the ATM to focus to each point across the distal facet of the fibre, and the output modes are the actual fields generated at the output of the fibre, which will be well formed foci only at positions within the isoplanatic patch. Here matrix $\mathbf{L}$ reduces the resolution of the output pixel basis to that commensurate with the size of the diffraction limited focus. Thus $\mathbf{S}$ natively describes the action of focussing using the ATM, and tends to the identity matrix as $\mathbf{D}$ tends to a perfectly diagonal matrix. The approximation that the normalised traces are equivalent in the first line of Eq. (19) is made under the assumption that the off-diagonal terms of $\mathbf{D}$ are small. An alternate way of expressing the fraction of power on the diagonal of $\mathbf{S}$ is given on the second line of Eq. (19), which corresponds to the mean power-ratio of foci created at the output of the fibre. This mean power-ratio also provides an estimate of the area of the isoplanatic patch $A_{iso}$ as a fraction of the total area of the core (third line), by considering the most compact form function $p_r(r, \theta)$ can take: as we expect $p_r$ to be highest near the guide-star, and the maximum value of $p_r$ is limited to 1, then the area over which this maximum region could extend gives an approximation to $A_{iso}$.

## Data availability
The experimental measurements shown in Figs. 4 and 5, and Supplementary Figs. 4–7, are available at https://doi.org/10.24378/exe.3203. All other data are available from the corresponding authors on request.

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

## Acknowledgements
D.B.P. thanks Dr. Ben Sherlock for help with early parts of the experiment, Prof. Jacopo Bertolotti for useful discussions and careful reading of the manuscript, and Emily Beth Wai-See for encouragement. D.B.P. and S.L. thank Dr. Sergey Turtaev and Dr. Ivo Leite for advice on setting up the experiment. S.L. acknowledges support from the National Natural Science Foundation of China under Grant no. 61705073. S.A.R.H. acknowledges financial support from a Royal Society TATA University Research Fellowship (RPG-2016-186). T.C. thanks the European Regional Development Fund (CZ.02.1.01/0.0/0.0/15_003/0000476) and the European Research Council (724530) for financial support. D.B.P. acknowledges financial support from the Royal Academy of Engineering and the European Research Council (804626).

## Author contributions
D.B.P., T.T. and T.C. conceived the idea for the project. D.B.P. supervised the project. S.A.R.H. and D.B.P. formulated the general memory effect theory. S.A.R.H., T.T. and D.B.P. investigated the quasi-radial memory effect. D.B.P., S.A.R.H., T.T. and S.L. performed the simulations. S.L. built the optical setup and performed all experiments and analysis, with support from D.B.P. and T.C. D.B.P., S.A.R.H. and S.L. wrote the manuscript with editorial input from all other authors.

## Competing interests
The authors declare no competing interests.
