## [Peer Review File · Nature Communications]

REVIEWER COMMENTS

Reviewer #1 (Remarks to the Author):

The manuscript “Memory effect assisted imaging through multimode optical fibres” by S. Li et al. reports on a new technique to characterise and partially image through MMFs, only requiring access to their proximal end. It consists in exploiting the memory effect correlations offered by the MMF in conjunction with guide-star assisted wavefront shaping. More precisely, the authors show that if one knows the incident field to focus on a single guide-star (such as a fluorescent bead) placed at the distal facet of the fibre, correlations of the transmitted optical field can be used to estimate an approximate transmission matrix (ATM) describing the scattering of light by the MMF. The ATM allows in particular to predict the proximal field modulations required to move the focus both azimuthally and radially as the conventional transmission matrix but is limited to a restricted region (a single isoplanatic patch). However the technique is more suitable for imaging inside complex environments, since it does not require any physical access to the distal facet of the fiber.

Overall, the manuscript is very well written and thorough. The method is novel and has exciting potential consequences. I think the work will be of great interest to the readers of Nature Communications and I am happy to recommend its publication.

Here are some comments that the authors may want to consider addressing before publication.

The first part of the manuscript presents a general framework describing how the memory effect translates into the transmission matrix correlations. In presence of memory effect the transmission matrix is actually a diagonal matrix in a well-chosen basis. Even if this mathematical analysis is interesting and may be mentioned as a starting point in the manuscript (and detailed in Supplementary), I would not go into the details for thin scattering media since it has already been deeply studied, like in ref 12 for the translational memory effect. In contrast, I would insist more on the azimuthal and quasi-radial memory effect present in MMFs which are at the heart of the paper. In this perspective, I recommend the authors to give additional details about the properties of these two types of memory effects. For instance for thin scattering media, the memory effect angle (for tilt/tilt memory effect) is proportional to λL with L thickness of the medium. Can the authors provide a similar analysis about the extension of the memory effects in fibres and show how the fibre specifications (core diameter, NA, length...) affect it? Additionally, the position of the focal spot also changes the size of the patch both azimuthally and radially. Can the authors discuss these points, maybe by providing a more detailed analysis of Figure 5?

As stated in the manuscript “The lack of a true ‘radially shifting’ memory effect [does not permit to] perform imaging based on 2D scanning of these speckle patterns as in ref [20]”. The strategy of the authors is not to raster scan the focus but more interestingly, to reconstruct an approximate transmission matrix (ATM) from a single row of ugs using the memory effect correlations and assuming the TM is unitary and diagonal in the PIM basis. The key point here is to find the correct PIM basis. Can the authors give more details on how they proceed to estimate it? As shown in Figure 3, the error directly affects the size of the overall memory effect patch. Would it be possible to improve the estimation by maximizing p_d numerically while testing different basis? Is p_d the only metric to characterise the fidelity of the ATM? What about the width of the diagonal?

If the isoplanatic patch does not reach across the entire output facet of the fibre, what are the possibilities to enlarge it? Measuring another ugs at a different spatial position, by

moving the guide-star or even playing with several guide-stars at the same time. Would that be a solution?

Experimentally, what happens if the fibre is twisted, does the axial symmetry at the origin of this memory effect type can still be used? Do the authors expect the same impact on the imaging capabilities as bending the fiber in Figure S7?

Also, the reconstruction of the ATM requires a calibration step, specifically the knowledge of ugs. Here the authors use a single camera pixel placed at the distal facet of the fibre to estimate it. This makes the experiment easier than using for example a fluorescent reporter. How the authors would proceed in a real situation to perform this calibration step while being sure of measuring the “correct” transmission matrix?

The authors mentioned the use of a quadratic phase term to move the focus axially. Is this can be used to reconstruct a "3D ATM" that would enable 3D imaging at the distal facet of the fibre?

Minor suggestions:

* SI§10 “position” should be “position”

Reviewer #2 (Remarks to the Author):

Memory effect assisted imaging through multimode optical fibres

The manuscript explains how to use first order correlations (memory effect), and a priori information of the fiber under test (need to know the basis on which the transmission matrix (TM) is diagonal) to retrieve partially the transmission matrix of the fiber. The authors demonstrate how to use this approximate TM (ATM) to focus light through the fiber on a guide-star, and for imaging. The authors also provide a comparison with the fully sampled TM of the fiber.

The manuscript reads globally very well, and the idea is original. The rotational memory effect in fiber was already demonstrated few years ago (Amitonova et al., Opt. Exp 2015, ref 35 of this manuscript). The authors are here using this effect for imaging, and using the ATM to exploit a quasi radial memory effect for point scanning at the tip of a fiber in a clever way. The demonstration of the memory effects with a matrix approach is very elegant and clear. The supplementary videos are clear too. I have some questions that need clarification prior to publication:

- What could be the effects of long-range correlations in the estimate of the transmission matrix? (As in “Long-range spatio-temporal correlations in multimode fibers for pulse delivery”, Wen Xiong, Chia Wei Hsu & Hui Cao, Nature Communications volume 10, Article number: 2973 (2019))

- What could help getting a better estimate of the basis in which the TM is diagonal? In page 8, you specify that 15% of the TM power is on its diagonal. How can we boost this number? It looks like this number is crucial to get a good isoplanetic patch. Is this because of the

imperfection of the fiber (the core cladding interface not being smooth, or imperfection of the refractive index profile within the core) itself? Could an improved alignment/ correction of the optical aberrations using the DMD help too?

- Could this method be extended to the measurement of spectrally resolved and temporally resolved TM (as in "Control of the temporal and polarization response of a multimode fiber", Mickael Mounaix & Joel Carpenter, Nature Communications volume 10, Article number: 5085 (2019))? That would be helpful to enhance non-linear interactions at the tip of a fiber, or for pulse delivery.

- Why are the authors not measuring the output field using off axis holography instead of phase stepping holography? The point of using a DMD is having a faster speed, so why measuring 4 frames with different phases (from your phase stepping approach) rather than a single shot off-axis? Is there a specific interest into phase-stepping?

- How would you handle multiple guide stars (different spatial positions, could be coherent or incoherent, like two fluorescent beads imbedded in a sample) to retrieve the ATM? Would it help to have multiple guide stars in general?

- With a single guide-star, how accurate could you place the guide-star (real or artificial) in a future "in vivo" experiment if using the experimental setup as an endoscope (as mentioned by authors in page 9 right column)? The location of the guide star looks crucial as it determines the isoplanetic patch (Figure 5).

- There is no mention of the loss of the system (mode dependent loss, insertion loss, DMD loss etc). What is the total loss?

- The naming "fully sampled TM" could be confusing, as only a quarter of the full transmission matrix is being measured (one polarization in, one polarization out)

- Could the ATM be used for point spread function engineering, as mentioned in supplementary section S5? (similarly to "Transmission-matrix-based point-spread-function engineering through a complex medium", Boniface et al. Optica Vol. 4, Issue 1, pp. 54-59 (2017))

Also some minor questions:

- When you describe the memory effect in arbitrary geometry, could your method/formalism be applied to relatively thin scattering media (where you can't measure all the modes of the system)? Not all disordered media satisfy the constraint "the TM is unitary" that you explain in page 3.

- Can the method/formalism be applied to graded index fibre too? I imagine the PIM basis would be different to the case of a step index fibre, but it should exist too right?

- Can the method be applied to a multimode fiber of any length (even >km length)? How the loss (mode dependent loss and insertion loss, also absorption etc) can affect the estimate of the TM?

- In the case of the shift-shift memory effect, $M = \text{diag}[e^{-i(k_x dx + k_y dy)}]$ (page 3 left column). Why do you have "-i" and not "i" as in the tilt tilt memory effect?

- The sentence "P is sufficiently oversampled in real-space such that $P_{\text{dagger}}P \sim I$." in page 3 right column is not clear. Why is that?

- I would recommend adding a sentence in the caption of Figure 3 for clarification. If a reader in a hurry only looks at the caption and reads the captions, it would be hard to understand the message of Figure 3 without reading the main text. Maybe a sentence that explains that the power-ratio relates to the spot-scanning efficiency?

- I would add the sample location in the experimental apparatus (Fig 4), especially because you have the imaging Figures just next to it.

- What is the transport mean free path of the studied step index fibre (as mentioned in page 10 left column)?

Some comments on the Supplementary document:

- In the paragraph S3 (experimental setup), I would recommend changing the "master/slave" naming for the DMD and camera to an alternative.

- In Equation S18, is it T^{-1} or T_{dagger} that is being calculated? While P is unitary (as it seems that $PP_{\text{dagger}} = \text{Identity}$ from equation S8), is D not necessarily unitary?

- In Fig S5 (and paragraphs S5 and S6), I would recommend calculating the signal to background ratio (SBR) of focusing for the single spot focusing, as it is a metric commonly used for probing the quality of the achieved focused spot. (as in Vellekoop and Mosk, Optics Letters 2007). I would recommend mentioning the SBR values in the main document too.

- In Figure S7, it is hard to see a change of contrast upon bending. I would recommend adding the signal to background ratio of focusing in each case.

- There is a small typo in the last paragraph S10 (poistion instead of position)

Reply to Reviewer Comments for “Memory effect assisted imaging through multimode optical fibres”

We thank the reviewers for taking the time to study our manuscript in detail, and thank them for their supportive comments about our work. We agree with the vast majority of the suggested changes which we think have strengthened and improved the clarity of our manuscript. Below we reply to each comment on a point-by-point basis in green, and highlight where the text has been changed.

Reviewer #1 (Remarks to the Author):

The manuscript “Memory effect assisted imaging through multimode optical fibres” by S. Li et al. reports on a new technique to characterise and partially image through MMFs, only requiring access to their proximal end. It consists in exploiting the memory effect correlations offered by the MMF in conjunction with guide-star assisted wavefront shaping. More precisely, the authors show that if one knows the incident field to focus on a single guide-star (such as a fluorescent bead) placed at the distal facet of the fibre, correlations of the transmitted optical field can be used to estimate an approximate transmission matrix (ATM) describing the scattering of light by the MMF. The ATM allows in particular to predict the proximal field modulations required to move the focus both azimuthally and radially as the conventional transmission matrix but is limited to a restricted region (a single isoplanatic patch). However the technique is more suitable for imaging inside complex environments, since it does not require any physical access to the distal facet of the fiber. Overall, the manuscript is very well written and thorough. The method is novel and has exciting potential consequences. I think the work will be of great interest to the readers of Nature Communications and I am happy to recommend its publication.

We thank the reviewer for their positive comments about our work.

Here are some comments that the authors may want to consider addressing before publication.

(1.1) The first part of the manuscript presents a general framework describing how the memory effect translates into the transmission matrix correlations. In presence of memory effect the transmission matrix is actually a diagonal matrix in a well-chosen basis. Even if this mathematical analysis is interesting and may be mentioned as a starting point in the manuscript (and detailed in Supplementary), I would not go into the details for thin scattering media since it has already been deeply studied, like in ref 12 for the translational memory effect.

We understand the reviewer’s point-of-view here, but we would prefer to leave the introduction as it is for the following reasons:

- The description of the tilt-tilt memory effect is used as a familiar example to describe how the framework of Equation 1 may be applied.
- We thought about moving the detailed description of the shift—shift memory effect to the supplementary material, however it links to several other ideas in the paper: the rotational memory effect in a fibre is a rotational analogue of this shift—shift memory effect, and this passage also introduces the concept of the transport mean free path which is necessary to understand the transport mean free path of our experimentally measured fibre.

As Nature Communications is targeted towards the more general reader, we hope the reviewer agrees with our decision to leave this introductory material in the main paper.

(1.2) In contrast, I would insist more on the azimuthal and quasi-radial memory effect present in MMFs which are at the heart of the paper. In this perspective, I recommend the authors to give additional details about the properties of these two types of memory effects.

For instance for thin scattering media, the memory effect angle (for tilt/tilt memory effect) is proportional to λL with L thickness of the medium. Can the authors provide a similar analysis about the extension of the memory effects in fibres and show how the fibre specifications (core diameter, NA, length...) affect it?

We agree with the reviewer that it would be interesting to provide more analysis about the limitations on the rotational and quasi-radial memory effects in MMFs, and we have now attempted to do so. In the case of the tilt-tilt memory effect, the memory effect angle θ is maximised (i.e. $\theta_{\max} = \pi/2$) in the limit of the scattering layer tending to zero thickness. When considering the tilt-tilt memory effect, a real scattering layer can be parameterised by two numbers capturing how it departs from this ideal case: the scattering mean free path (l_s), and the thickness L . Under the assumption that $L \gg l_s$ it is possible to describe how the maximum memory effect angle reduces as a function of L , which as the reviewer notes is given by: $\theta_{\max} \propto \lambda/L$. Considering the situation in the TM framework, we can consider L as a single parameter that determines the degree of off-diagonal spreading of the TM in the real-space basis. It is also important to note that conducting the same experiments with the scattering medium translated, or tilted a small amount, would lead in average to the same effects, therefore alignment of the medium with respect to the probing optics does not play any important role.

To attempt to perform a similar analysis in the case of a MMF, we first identify the case when the rotational/quasi radial memory effects are maximised (i.e. $\theta_{\text{rotational}} = 2\pi$, $p_{\text{radial}} = r$). This occurs for a straight cylindrically symmetric fibre with a perfectly known refractive index profile (in our case a step profile). In this ideal case, the rotational and quasi-radial memory effects are maximised for any core radius a , numerical aperture (NA) and length L . We note that the standard way we calculate the fibre modes itself contains some approximations that deteriorate with increasing NA and length - but this could be rectified by using a more sophisticated treatment to find fibre modes at higher NA.

So, what causes a real fibre to depart from the ideal case, and for power to be found in off-diagonal elements in the PIM basis? This was the subject of an earlier study by some of the authors of the present manuscript (ref. [33]). They showed that contributing factors are:

- (i) Misalignment of the inputs and outputs of the fibre, i.e. small misalignments in the position of the fibre tips in 3D (x , y , z - here taken to be parallel to the fibre axis), and tilts in 2D (tilt about x , tilt about y), and defocus. This constitutes 6 parameters at each end, therefore 12 parameters in total.
- (ii) Departure of the refractive index profile from the ideal (and assumed) case. For example, during manufacture, undesired diffusion of the dopants used to create the step in refractive index leads to a departure in the refractive index profile from the ideal one, thus changing the structure of the PIMs supported in the fibre.
- (iii) Bending/twisting of the fibre also changes the mode structure.

All of these factors result in power spreading into the off-diagonal elements when the TM is expressed in the ideal PIM basis. For short lengths of step index MMF of up to $\sim 30\text{cm}$,

input/output misalignment is the main contributing factor - and that the TM was highly sensitive to even very minor misalignments. In contrast to the case of the tilt-tilt memory effect, the large number of degrees-of-freedom involved render it challenging to analytically describe how the rotational/quasi-radial memory effect depend on these parameters. Nevertheless, it is possible to make a qualitative statement about how the size of the isoplanatic patch is related to the fraction of power on the diagonal of the full TM. We have now added a new paragraph discussing the above in the main paper, along with a new derivation in the Methods.

In the main paper we now say:

“The angular extent of the tilt–tilt memory effect, α , is related to a single parameter describing the sample: the thickness of the scattering layer L_s , so that $\alpha \sim \lambda/(2\pi L_s)$, where λ is the incident wavelength [13]. This raises the question of whether it is also possible to derive a similar equation describing the limits of the memory effects in MMFs. We start by noting that, as shown in Fig. 3a, an ideal perfectly straight MMF supports a rotational memory effect of 2π , and a quasi-radial memory effect of r , regardless of its numerical aperture, core diameter or length. However, the size and shape of the isoplanatic patch exhibited by real MMFs is highly sensitive to a large number of parameters. These include misalignments of the optical system at the input and output of the fibre (at least 6-degrees of freedom at each end), the departure of the cross-sectional refractive index profile of the fibre from the ideal (or assumed) case, how this profile varies along the fibre’s length, and the bend configuration of the fibre [33]. This renders finding an analytical expression for the extent of the memory effect in MMFs a challenging task. Nevertheless, the combined effect of these parameters is to spread power into the off-diagonal elements of the real TM D , and we find that the approximate area of the isoplanatic patch, A_{iso} , is related to the fraction of power remaining on the diagonal of the actual TM, p_d , according to: $A_{iso} \sim \pi r^2 p_d$. See Methods for a derivation of this expression and the approximations used. This trend is qualitatively apparent in Fig. (3).”

(1.3) Additionally, the position of the focal spot also changes the size of the patch both azimuthally and radially. Can the authors discuss these points, maybe by providing a more detailed analysis of Figure 5?

We agree with the reviewer that the article would benefit from a more detailed analysis of Fig 5. We now say the following:

“It is straight forward to test alternative guide-star positions in our proof-of-principle experiments - moving the guide-star simply means choosing a different camera pixel to use for ATM construction. Figure (5) shows the experimental power ratio maps for three different choices of guide-star location, along with the theoretical level of overlap with the PIMs in each case. As expected, the area of the isoplanatic patch is maximised when the guide-star is placed on the core-cladding boundary (Fig. (5a)), enabling recovery of most of the diagonal elements of the ATM in the PIM basis. As the guide-star location is moved radially inwards towards the centre of the core, the number of PIMs sampled by the guide-star progressively decreases (Figs. (5a-c)). When the guide-star is placed at a radius of $a/2$, we observe a strong arc in the isoplanatic patch, a signature of the rotational memory effect (Fig. (5e)). This isoplanatic arc is formed as projection of foci to the same radius as the guide-star require the same combination of PIMs, the phase delays of which have been accurately sampled in the ATM. However we also observe a sharp reduction in the power-ratio when the focus is moved radially in Fig. (5e) as these points require the

constructive interference of different combinations of PIMs, some of which are not well-sampled by the guide-star. When the guide-star is located on the fibre axis, it exclusively samples only the PIMs with a vortex charge of $l = 0$ (Fig. (5c)), and the isoplanatic patch contracts to a single spot as shown in Fig. (5f).”

(1.4) As stated in the manuscript “The lack of a true ‘radially shifting’ memory effect [does not permit to] perform imaging based on 2D scanning of these speckle patterns as in ref [20]”. The strategy of the authors is not to raster scan the focus but more interestingly, to reconstruct an approximate transmission matrix (ATM) from a single row ugs using the memory effect correlations and assuming the TM is unitary and diagonal in the PIM basis. The key point here is to find the correct PIM basis. Can the authors give more details on how they proceed to estimate it?

It has previously been shown that a short length of MMF will have a TM that is highly diagonal in the PIM basis (see ref [33]). This basis can be found by optimising over misalignment parameters at the inputs and outputs, and optimising the parameters of the fibre itself. Here, rather than performing such a detailed optimisation (which we note in the conclusion would facilitate an increase in the size of the isoplanatic patch), in this work we followed some steps to achieve a *coarse* alignment that are also detailed in ref [33]. We now explain this in the Methods section where we say:

“To account for some of the experimental misalignments, misalignment operators are measured and applied at each end of the fibre that remove course position and tilt misalignments of the input and output. These operators are found following the coarse alignment methods in ref. [33]. In brief, we make use of the fact that both ends of MMF are accessible before it is employed as an endoscope, and measure the full real-space TM of the fibre (at a single polarisation). This real space TM can be processed *without* optimisation to reveal estimates of the position and tilt misalignments of the inputs and outputs in the following ways: (i) The central position of the core at output can be found by measuring the centre-of-mass of the sum of the intensity of all output measurements. (ii) The tilt of the output can be found by measuring the centre-of-mass of the sum of the intensity of the Fourier transform of all output measurements. Both of these methods can be understood as operations involving summing over the absolute square of the columns of the measured TM. Under the assumption that the TM is unitary, the transpose of this TM is equivalent to the TM if it were measured in the opposite direction through the fibre. Therefore the position of the core at the input, and the tilt of the input light can be found by performing the equivalent to (i) and (ii), but summing over the rows rather than the columns of the TM. Once these misalignments are found, they can be represented as misalignment matrices at the proximal end (R_{pr}) and at the distal end (R_{dl}), as described in more detail in SI §S4.”

(1.5) As shown in Figure 3, the error directly affects the size of the overall memory effect patch. Would it be possible to improve the estimation by maximizing p_d numerically while testing different basis? Is p_d the only metric to characterise the fidelity of the ATM? What about the width of the diagonal?

In the present work, estimation of PIM basis has been done purely on the grounds of fibre parameters given by the fibre’s manufacturer, along with some coarse misalignment correction as described in our answer to question (1.4) above. Indeed the suggestion that the referee describes to improve the estimation of the PIM basis is closely related to some of the authors previous studies [ref 33], where it was shown that optimisation of the fibre parameters (radial refractive index profile) along with fine tuning of input/output

misalignments and aberrations can be used to identify a PIM basis with the majority of the power appearing on the diagonal. In the present work we focused on demonstrating the proof-of-principle of memory effect assisted guide-star imaging, and in future work we plan to explore this possibility in conjunction with more detailed optimisation to find the PIM basis, as the reviewer suggests. We mention this in the conclusion of the current manuscript where we say:

“In our proof of principle experiments, the isoplanatic patch does not extend over the full area of the distal facet. However our simulations show (see Fig. (3)) that the area of this isoplanatic patch can be significantly increased by making a better estimate of the basis in which the TM of the MMF is diagonal. Such an estimate can be achieved by optimising the fibre parameters with access to the full TM, before subsequent deployment of the system as a flexible micro-endoscope [33,46], which will be a focus of our efforts in future work.”

We have also added a reference to a recent paper that is relevant to this idea (now ref 46): Matthès, M. W., Bromberg, Y., de Rosny, J., & Popoff, S. M. (2020). *Learning and avoiding disorder in multimode fibers*. arXiv: 2010.14813.

In answer to the second part of the question: We think the reviewer is correct that there are other criteria that one could use, in addition/as an alternative to p_d , to characterise how well the diagonal basis for the TM has been estimated. Choices could include, for example, the total sparsity of the TM in the quasi-diagonal basis (i.e. how many elements contain near zero power), or how unitary the TM is measured to be. Combinations of such criteria could be used as an objective function to optimise the estimated diagonal basis.

(1.6) If the isoplanatic patch does not reach across the entire output facet of the fibre, what are the possibilities to enlarge it? Measuring another ugs at a different spatial position, by moving the guide-star or even playing with several guide-stars at the same time. Would that be a solution?

Firstly, we think it may be difficult to remotely move a guide-star around the distal facet in a controllable way. However the reviewer suggests an interesting alternative possibility that we are currently actively exploring in ongoing work: increasing the area of the isoplanatic patch by using multiple guide-stars mounted on the distal facet of the fibre. We mention this prospect in the conclusion to the current manuscript where we say:

“To image through longer MMFs which possess higher levels of mode coupling (and therefore lower values of p_d even after fibre parameter optimisation), we envisage that multiple guide-stars, distinguished by, for example, emission frequency or speckle pattern contrast [47,48], could be used to increase the total area of the distal facet through which it is possible to image.”

We have also added a reference to a recent paper that is also relevant to this idea (now ref 48): Boniface, A., Dong, J., & Gigan, S. (2020). *Non-invasive focusing and imaging in scattering media with a fluorescence-based transmission matrix*. Nature Communications, 11, 6154 (2020).

(1.7) Experimentally, what happens if the fibre is twisted, does the axial symmetry at the origin of this memory effect type can still be used? Do the authors expect the same impact on the imaging capabilities as bending the fiber in Figure S7?

We think that the answer to this question depends upon the degree of the twist. A small amount of twisting of a bent fibre has also been studied in ref [33] in which the TM survived very well: twisting added an additional phase to the PIMs dependent upon their azimuthal index ℓ , which resulted in the whole image rotating as the fibre was twisted. In this case we expect our guide-star assisted imaging method to enable measurement of these relative phase shifts and thus restore the original orientation of the image. For higher degrees of twist, at some point the geometry and the material properties of the fibre unavoidably change, which will likely impact the TM more severely and disrupt imaging. Initially we expect this to mainly influence the propagation constants of the PIMs (albeit in a more complicated way than simply scaling with azimuthal index as above), hence repeating the guide-star measurements should indeed help to restore the imaging performance in an isoplanatic patch around the guide-star. As we have not tested this scenario in our current work, we have elected not to add a detailed discussion of this to the manuscript.

(1.8) Also, the reconstruction of the ATM requires a calibration step, specifically the knowledge of u_{gs} . Here the authors use a single camera pixel placed at the distal facet of the fibre to estimate it. This makes the experiment easier than using for example a fluorescent reporter. How the authors would proceed in a real situation to perform this calibration step while being sure of measuring the “correct” transmission matrix?

In the case of a real guide-star placed at the end of the fibre, it would be possible to optimise the input field to focus onto a fluorescent guide-star by measuring total emitted fluorescence transmitted back to the proximal end as the input was modulated, thus finding u_{gs} . This assumes that the fluorescence is emitted only by the guide-star. This optimisation could be conducted using phase stepping holography as detailed in ref [4], in $3N$ measurements (where N is the number of modes of a single polarisation supported by the fibre). SI 8 describes this possibility in more detail.

(1.9) The authors mentioned the use of a quadratic phase term to move the focus axially. Is this can be used to reconstruct a "3D ATM" that would enable 3D imaging at the distal facet of the fibre?

The TM is natively a 2D object: it describes the transformation of light on a 2D plane at the input of the fibre to a second 2D plane at the output of the fibre. If there is free-space at the distal end of the fibre, it is possible to use this 2D TM to scan the focus in 3D at the end of the fibre - in the case of a cylindrically symmetric fibre, axial refocussing of the output focus is achieved by simply adding a quadratic phase term to the input SLM. This is a useful and computationally straight-forward simplification originating from the conservation of propagation constants of the modes. Describing this situation by adding an additional dimension to the TM (i.e. creating a 3D TM) is in principle not necessary, since outside the fibre light transport follows simple free-space propagation, and so all the information of the 3D light-field is held by a single 2D plane (and connected to every other plane around it in free-space by the wave equation). Therefore the 2D TM alone can be used to focus any distance from the fibre.

Minor suggestions:

* Sli_z^{1/2}10 “position” should be “position”

We thank the reviewer for noticing this and have now corrected it.

Reviewer #2 (Remarks to the Author):

Memory effect assisted imaging through multimode optical fibres

The manuscript explains how to use first order correlations (memory effect), and a priori information of the fiber under test (need to know the basis on which the transmission matrix (TM) is diagonal) to retrieve partially the transmission matrix of the fiber. The authors demonstrate how to use this approximate TM (ATM) to focus light through the fiber on a guide-star, and for imaging. The authors also provide a comparison with the fully sampled TM of the fiber.

The manuscript reads globally very well, and the idea is original. The rotational memory effect in fiber was already demonstrated few years ago (Amitonova et al., Opt. Exp 2015, ref 35 of this manuscript). The authors are here using this effect for imaging, and using the ATM to exploit a quasi radial memory effect for point scanning at the tip of a fiber in a clever way. The demonstration of the memory effects with a matrix approach is very elegant and clear. The supplementary videos are clear too.

We thank the reviewer for their encouraging comments about our work.

I have some questions that need clarification prior to publication:

(2.1) What could be the effects of long-range correlations in the estimate of the transmission matrix? (As in “Long-range spatio-temporal correlations in multimode fibers for pulse delivery”, Wen Xiong, Chia Wei Hsu & Hui Cao, Nature Communications volume 10, Article number: 2973 (2019))

Our work relies on being able to predict a basis in which the TM is quasi-diagonal with minimal coupling. In contrast, the interesting paper mentioned by the reviewer exploits strong coupling in the TM - i.e the fibre is tightly bent to induce a large degree of mode-coupling, deliberately destroying the properties of the PIMs that we make use of in our work. The authors then cleverly take advantage of this coupling by uncovering spatio-temporal correlations that emerge when the input light explores a range of propagation constants. Therefore, this work studies a fundamentally different system from the weakly coupled model our work is based on. However, it is an interesting example of correlations being usefully exploited in a fibre and so we now cite it as ref 55 in the conclusion of our paper where we say:

“In addition to spatial correlations in weakly coupled MMFs, we also note that spatio-temporal correlations have been identified in MMFs with strong mode coupling that may prove useful for pulse delivery [55,56].”

(2.2) What could help getting a better estimate of the basis in which the TM is diagonal? In page 8, you specify that 15% of the TM power is on its diagonal. How can we boost this number? It looks like this number is crucial to get a good isoplanetic patch. Is this because of the imperfection of the fiber (the core cladding interface not being smooth, or imperfection of the refractive index profile within the core) itself? Could an improved alignment/ correction of the optical aberrations using the DMD help too?

In the present work we measure and correct for aberrations from the DMD and other parts of the optical system around the fibre in a pre-calibration step (see SI section 4), largely removing this effect from the fibre TM. In the current paper, estimation of PIM basis was based purely on the grounds of fibre parameters given by the fibre's manufacturer, along with some coarse misalignment correction as described in our answer to reviewer 1 question (1.4) above. The reviewer is correct that to boost this number we would need to measure and remove the remaining input and output misalignments and aberrations, and also fine tune the parameters of the fibre itself. Previous work by some of the authors [ref 33], showed that optimisation of the fibre parameters (particularly its radial refractive index profile), along with fine tuning of input/output misalignments can be used to identify a PIM basis with the majority of the power appearing on the diagonal, for a fibre of similar length. In the present work we focused on demonstrating the proof-of-principle of memory effect assisted guide-star imaging, and in future work we plan to explore this possibility in conjunction with more detailed optimisation to find the PIM basis. We mention this in the conclusion of the current manuscript where we say:

“In our proof of principle experiments, the isoplanatic patch does not extend over the full area of the distal facet. However our simulations show (see Fig. (3)) that the area of this isoplanatic patch can be significantly increased by making a better estimate of the basis in which the TM of the MMF is diagonal. Such an estimate can be achieved by optimising the fibre parameters with access to the full TM, before subsequent deployment of the system as a flexible micro-endoscope [33,46], which will be a focus of our efforts in future work.”

We have also added a reference to a recent paper that is relevant to this idea (now ref 46): Matthès, M. W., Bromberg, Y., de Rosny, J., & Popoff, S. M. (2020). Learning and avoiding disorder in multimode fibers. arXiv: 2010.14813.

(2.3) Could this method be extended to the measurement of spectrally resolved and temporally resolved TM (as in “Control of the temporal and polarization response of a multimode fiber”, Mickael Mounaix & Joel Carpenter, Nature Communications volume 10, Article number: 5085 (2019))? That would be helpful to enhance non-linear interactions at the tip of a fiber, or for pulse delivery.

We first note that since circular polarisation coupling is minimal in the short piece of fibre used in our study (see also ref [33]), then in principle the ATM at two orthogonal polarisations could be independently measured. Considering spectrally resolved measurements, then if a fluorescent guide-star possesses a broad excitation bandwidth, then the ATM at a series of wavelengths within that excitation bandwidth could be independently measured. Therefore we think that the resulting multispectral TM could enable spatiotemporal focussing at the output. However, there is an open question about whether the relative phase between the spectral components could be recovered from guide-star measurements, which is needed for full temporal control of the output field. Therefore, as this concept is speculative, we have elected not to mention this possibility in the main paper. Nonetheless we thank the reviewer for drawing our attention to the interesting highlighted paper which we now cite (ref 56) as an example of pulse delivery in MMFs.

(2.4) Why are the authors not measuring the output field using off axis holography instead of phase stepping holography? The point of using a DMD is having a faster speed, so why measuring 4 frames with different phases (from your phase stepping approach) rather than a single shot off-axis? Is there a specific interest into phase-stepping?

The guide star provides an intensity response to its illumination at single location. In our work this guide-star is emulated by a single CCD pixel. Single shot off-axis holography requires the measurement of the interference pattern at multiple locations (i.e. over multiple camera pixels), enabling processing of this image by a 2D spatial Fourier transformation to select the off-axis component of the field and separate it from the DC and other components. However, we are aiming to emulate the situation of having a single guide-star at the output facet, where there is no opportunity to record the field at multiple pixels. Therefore in the guide-star case, which our paper is focussed on, phase stepping holography is the only viable option.

(2.5) How would you handle multiple guide stars (different spatial positions, could be coherent or incoherent, like two fluorescent beads imbedded in a sample) to retrieve the ATM? Would it help to have multiple guide stars in general?

We think multiple guide-stars could be used to increase the size of the isoplanatic patch. If the guide-stars emit at different frequencies then it would be possible to use spectral discrimination to separately focus on each one, and thus image in an isoplanatic patch around each one. If the guide-stars emit incoherently at the same frequency (such as fluorescent beads), then it may be possible to use the methods in refs 47 or 48 to separately focus on each guide-star. We briefly discuss this in the conclusion of the paper where we say:

“..we envisage that multiple guide-stars, distinguished by, for example, emission frequency or speckle pattern contrast [47,48], could be used to increase the total area of the distal facet through which it is possible to image.”

(2.6) With a single guide-star, how accurate could you place the guide-star (real or artificial) in a future “in vivo” experiment if using the experimental setup as an endoscope (as mentioned by authors in page 9 right column)? The location of the guide star looks crucial as it determines the isoplanetic patch (Figure 5).

As the reviewer notes, it is preferable to place a guide-star near the core-cladding boundary. However if using multiple guide-stars this placement is perhaps not so crucial and several guide-stars distributed at random positions over core may work well. Interestingly, a recent paper describes a way to controllably place fluorescent material on the end of a fibre:

Vikram Kamaljith, Michael G Tanner, Harry AC Wood, Kerrienne Harrington, Debaditya Choudhury, Mark Bradley, and Robert R Thomson. Ultrafast-laser-ablation-assisted spatially selective attachment of fluorescent sensors onto optical fibers. *Optics Letters*, 45(10):2716–2719, 2020.

and so we have added this reference to the conclusion where we now say:

“If necessary, accurate guide-star placement could potentially be achieved using, for example, laser-ablation-assisted attachment of fluorescent sensors [49].”

(2.7) There is no mention of the loss of the system (mode dependent loss, insertion loss, DMD loss etc). What is the total loss?

We agree that some discussion of the loss in our optical system would be informative. We now include an additional section (S9) in the supplementary information estimating the

losses in the various parts of the experimental system, which is pointed to in the main text where the experiment is introduced. S9 reads:

“Here we estimate the level of loss in the optical system.

DMD loss: This is the major source of loss in our optical system. We choose to use a DMD to shape input light fields as DMDs offer a high modulation rate (up to ~ 20 kHz) and have been shown to create fields of high fidelity (see ref [38]). However, the binary amplitude modulation of DMDs means that these advantages come at the expense of low light conversion efficiency. Accounting for diffraction from the pixels themselves (which create multiple additional diffraction orders), and the inherently lossy nature of amplitude and phase modulation, we estimate that $\sim 1\%$ of the incident power is transmitted into the target beam.

Insertion loss: The numerical aperture of the light focused into the fibre is set to slightly overfill the NA of the fibre, thus resulting in some insertion loss during the measurement of the TM. The fibre in our experiment is not antireflection coated, and so incident light, travelling from air medium to glass suffers an additional $\sim 4\%$ reflection loss on the input to the fibre. These losses could be mitigated by matching the NA of the incident light to that of the fibre during TM measurement and using an antireflection coated fibre.

Mode dependent loss: PIMs of higher mode indices possess more power propagating close to the critical angle of total internal reflection as they propagate through the MMF, and so are more likely to lose power into the cladding due to fibre imperfections. However as our intended micro-endoscope application uses short ~ 30 cm lengths of fibre, we expect this loss to be minimal. However, residual misalignments of the input will artificially increase the apparent mode dependent loss - as an inaccurately estimated PIM basis will result in inaccurately shaped modes launched into the fibre, and the modes of higher mode index lose extra light into the cladding. In the future this can be rectified by using optimization to precisely align the input to the fibre, as demonstrated in refs. [33, 46].”

(2.8) The naming “fully sampled TM” could be confusing, as only a quarter of the full transmission matrix is being measured (one polarization in, one polarization out)

We agree with the reviewer and have now added a clarification the first time we use this phrase. We also note that since the input circular polarization is maintained over propagation through short lengths of fibre, we sample an orthogonal sub-space of the full TM, which is sufficient for high-contrast imaging behind the distal end [33].

(2.9) Could the ATM be used for point spread function engineering, as mentioned in supplementary section S5? (similarly to “Transmission-matrix-based point-spread-function engineering through a complex medium”, Boniface et al. Optica Vol. 4, Issue 1, pp. 54-59 (2017))

Within the isoplanatic patch the ATM can be used in the same way as the fully sampled TM, and so point-spread function engineering should be possible. Indeed, the axial refocussing of spot position we carry out in order to image the resolution target in Fig 4 goes some way to demonstrate this. We now make this point in the paper where we cite the highlighted reference (as ref [42]), and say:

“More generally, these refocussing and pattern projection results show that point-spread-function engineering should be possible within the isoplanatic patch [42].”

Also some minor questions:

(2.10) When you describe the memory effect in arbitrary geometry, could your method/formalism be applied to relatively thin scattering media (where you can't measure all the modes of the system)? Not all disordered media satisfy the constraint “the TM is unitary” that you explain in page 3.

The reviewer raises an interesting point here that we are currently actively investigating in ongoing work. In brief, yes we think it may be possible to use a more elaborate method to estimate a non-unitary TM, namely, optimisation to estimate such a TM that both agrees with the guide-star measurements, and is sparse in the predicted basis (in this case the PIM basis). We have now extended our mention in this in the paper where we say:

“... points to a more sophisticated approach to reconstruct the ATM: instead of forcing D' to be diagonal, we could try to iteratively search for a non-diagonal and potentially non-unitary D' allowing the minimum power in off-diagonal elements [43], so that $PD'P^\dagger$ also perfectly focusses onto the guide-star when operating on input field ugs. This is a severely under-constrained problem that we plan to investigate in more detail in the future.”

(2.11) Can the method/formalism be applied to graded index fibre too? I imagine the PIM basis would be different to the case of a step index fibre, but it should exist too right?

We think this method could be applied to a graded index fibre too - as long as a model of the fibre was available enabling accurate prediction of the fibre modes. We note that a previous study by some of the authors (ref. [58]) showed that it is more difficult to predict the TM of a graded index fibre than in the step index case. Nonetheless [58] showed that using optimisation it was possible to make estimate a diagonal basis yielding 58% of the power on the diagonal. This corresponds to an isoplanatic patch covering ~half of the output facet of the fibre, with imaging capabilities similar to those shown in Fig 3(c,d,i,j).

Beyond this, we think that our method can apply to any system where it is possible to predict a basis in which the TM is quasi-diagonal. We emphasise this point in the final paragraph of the paper where we say the following:

“Our work provides a guide on how to efficiently hunt for and make use of memory effects in optical systems of arbitrary geometry. This concept may be applied to a range of other media including graded-index [57,58] and photonic crystal fibres [59], few-mode fibre bundles (which have the constraint that the guide-star is ideally placed in the far-field of the output [60]), photonic lanterns [61], opaque walls (i.e. imaging around corners) [62], and artificially engineered photonic networks and scattering systems.”

(2.12) Can the method be applied to a multimode fiber of any length (even >km length)? How the loss (mode dependent loss and insertion loss, also absorption etc) can affect the estimate of the TM?

In this work we have focussed on short lengths of fibre that are suitable for endoscopic imaging applications. Evidently longer fibres exhibit a greater level of mode coupling. Once the fibre length surpasses the transport mean-free-path (TMFP), the isoplanatic patch achievable using the ATM would shrink to a diffraction limited spot at the location of the guide-star. This places a limit on the length of fibre over which our method would work.

However, if a fibre is engineered to have a TMFP of over a km in length, then our method will work to some extent. Loss also renders the TM non-unitary - although this can potentially be accounted for as we discuss in answer to point 2.10 above.

(2.13) In the case of the shift-shift memory effect, $M = \text{diag}[e^{-i(k_x dx + k_y dy)}]$ (page 3 left column). Why do you have “-i” and not “i” as in the tilt tilt memory effect?

This is a subtle point: the shift-shift and tilt-tilt memory effects are Fourier duals of each other, one moving the output field, the other re-directing it. As a Fourier transform and its inverse contain factors of $\exp(-i k x)$ and $\exp(+i k x)$ respectively, the applied phases in the two memory effects must similarly differ by a minus sign. We now include a footnote clarifying this point.

(2.14) The sentence “P is sufficiently oversampled in real-space such that $P^{\dagger}P \sim I$.” in page 3 right column is not clear. Why is that?

The modes are continuous functions of position, and orthonormal with respect to an integral over all space. As the sampling in real space is increased, the matrix multiplication $P^{\dagger}P$ becomes an increasingly better approximation to the set of mode overlap integrals, and therefore ever closer to the identity matrix. We have now added a footnote to clarify this in the paper.

(2.15) I would recommend adding a sentence in the caption of Figure 3 for clarification. If a reader in a hurry only looks at the caption and reads the captions, it would be hard to understand the message of Figure 3 without reading the main text. Maybe a sentence that explains that the power-ratio relates to the spot-scanning efficiency?

Here we agree with the reviewer and have modified the caption to figure 3 as suggested. It now reads:

“The size and shape of the isoplanatic patch: (a-f) Simulated maps of the power-ratio of points focussed to different regions of the distal facet through non-ideal fibres with quasi-diagonal TM D. Higher values of the power-ratio indicate regions on the distal fibre facet where spot scanning can be achieved with greater contrast, leading to higher fidelity imaging. We see that the isoplanatic patch gradually shrinks around the location of the guide-star (marked by a red circle). Foci are created using ATM D' calculated from ugs using Eqn. (7). p_d , the power on the diagonal of D, decreases from panel (a) to (f) and is given at bottom of each panel. (g-l) Simulations of scanning imaging capabilities in each case.”

(2.16) I would add the sample location in the experimental apparatus (Fig 4), especially because you have the imaging Figures just next to it.

We agree with this suggestion and have now added a label indicating where the sample is inserted.

(2.17) What is the transport mean free path of the studied step index fibre (as mentioned in page 10 left column)?

We estimate the length of the fibre in our experiment to be a factor of ~ 0.02 of the transport mean free path in the PIM basis (i.e. the fibre length beyond which modal coupling is maximised, and the TM can be considered fully coupled). To make this

estimation we followed the methods described in *Principal modes in multi-mode fibers: exploring the crossover from weak to strong mode coupling*, by Wen Xiong, Philipp Ambichl, Yaron Bromberg, Brandon Redding, Stefan Rotter, and Hui Cao. *Optics Express*, 25(3):2709–2724, 2017 (now included as reference 39 in the paper).

We have now included this in our paper where we say:

“The degree of mode coupling in the TM can also be quantified by the ratio of L/l_f , where l_f is the transport mean-free-path (TMFP) in the fibre mode (PIM) basis, i.e. the estimated length of fibre beyond which the TM can be considered fully coupled. In this case our experimentally measured TM has a degree of mode coupling consistent with $L/l_f \sim 0.02$, a value we estimate by following the methods given in ref. [39].”

Some comments on the Supplementary document:

(2.18) In the paragraph S3 (experimental setup), I would recommend changing the “master/slave”) naming for the DMD and camera to an alternative.

We agree with the reviewer’s suggestion here, and have now removed this wording from the paper.

(2.19) In Equation S18, is it T^{-1} or T^\dagger that is being calculated? While P is unitary (as it seems that $PP^\dagger = \text{Identity}$ from equation S8), is D not necessarily unitary?

Here equation S18 defines the inverse of the unitary ATM T' rather than the inverse of the real potentially non-unitary TM T (notice it contains D' which is unitary). Therefore $T'^{-1} = T'^\dagger$.

(2.20) In Fig S5 (and paragraphs S5 and S6), I would recommend calculating the signal to background ratio (SBR) of focusing for the single spot focusing, as it is a metric commonly used for probing the quality of the achieved focused spot. (as in Vellekoop and Mosk, *Optics Letters* 2007). I would recommend mentioning the SBR values in the main document too.

Here we take a different view from the reviewer. We think the SBR is a more appropriate metric for assessing the quality of a focus in the regime where the number of spatial modes controlled by an SLM is lower than the number of modes supported in the scattering system. In that case, there is always some non-zero background level to compare to the intensity of the focus. However, in our experiment focussing through a MMF, the number of controllable modes exceeds the number of modes supported by the fibre, and so theoretically it could be possible to pump virtually all of the available power into the focus. In this case, the SBR would diverge to infinity and/or become very sensitive to camera noise. Therefore, in place of SBR, throughout this work we have elected to quantify the focussing efficiency with the *power-ratio* - which, as explained in the text, is defined as the ratio of the power found in the region of the focus, divided by the *total power* transmitted through the fibre (including the power in the point of interest). The power-ratio also shows the quality of the focussing, however does not diverge, but instead tends to 1 as all of the power is pumped into the target focus. We note that ref. [19] derives the relationship between the power-ratio and the SBR in the case of phase only modulation of input light with an SLM in the Fourier plane of the proximal facet.

(2.21) In Figure S7, it is hard to see a change of contrast upon bending. I would recommend adding the signal to background ratio of focusing in each case.

We agree that it would be useful for the reader to be able to more easily numerically quantify the change in power ratio, so we have now annotated each panel with the peak power ratio. As with the answer to the previous question, we have elected to continue using the power-ratio for the reasons given above.

(2.22) There is a small typo in the last paragraph S10 (poistion instead of position)

We thank the reviewer for noticing this error, which we have now corrected.

REVIEWER COMMENTS

Reviewer #1 (Remarks to the Author):

The authors have made a number of changes to referees comments and these are now mostly addressed either via correction or by a better explanation in the text. The manuscript can now be accepted for publication in Nature Communications.

Reviewer #2 (Remarks to the Author):

The authors have responded accurately to all the questions. The quality and clarity of the manuscript have improved. Hence I don't see further obstructions and I recommend the publication of the manuscript in Nature Communications.

Reply to Reviewer Comments for “Memory effect assisted imaging through multimode optical fibres”

We once again thank the reviewers for taking the time to study our manuscript in detail, and thank them for their supportive comments about our work. Below we reply to each comment on a point-by-point basis in green.

REVIEWERS' COMMENTS

Reviewer #1 (Remarks to the Author):

The authors have made a number of changes to referees comments and these are now mostly addressed either via correction or by a better explanation in the text. The manuscript can now be accepted for publication in Nature Communications.

We thank the reviewer for their supportive comments about our work.

Reviewer #2 (Remarks to the Author):

The authors have responded accurately to all the questions. The quality and clarity of the manuscript have improved. Hence I don't see further obstructions and I recommend the publication of the manuscript in Nature Communications.

We thank the reviewer for their supportive comments about our work.